*JCB* Journal of Cell Biology

# Aurora A promotes chromosome congression by activating the condensin-dependent pool of KIF4A

Elena Poser[1] , Renaud Caous[2] , Ulrike Gruneberg[2] , and Francis A. Barr[1]

Aurora kinases create phosphorylation gradients within the spindle during prometaphase and anaphase, thereby locally regulating factors that promote spindle organization, chromosome condensation and movement, and cytokinesis. We show that one such factor is the kinesin KIF4A, which is present along the chromosome axes throughout mitosis and the central spindle in anaphase. These two pools of KIF4A depend on condensin I and PRC1, respectively. Previous work has shown KIF4A is activated by Aurora B at the anaphase central spindle. However, whether or not chromosome-associated KIF4A bound to condensin I is regulated by Aurora kinases remain unclear. To determine the roles of the two different pools of KIF4A, we generated specific point mutants that are unable to interact with either condensin I or PRC1 or are deficient for Aurora kinase regulation. By analyzing these mutants, we show that Aurora A phosphorylates the condensin I–dependent pool of KIF4A and thus actively promotes chromosome congression from the spindle poles to the metaphase plate.

## Introduction

In prometaphase and metaphase, the two Aurora kinases A and B regulate chromosome congression by monitoring both chromosome position and the process of chromosome biorientation (Lampson and Cheeseman, 2011). Tension is generated across the kinetochores of bioriented chromosomes, enabling these to be discriminated from other attachment geometries by an error-correction pathway. This process is primarily regulated by centromeric Aurora B, which is physically close enough to the outer kinetochore to exert its effects only in the absence of tension, and thus provides a molecular readout of biorientation and tension (Lampson and Cheeseman, 2011; Wang et al., 2011). Phosphorylation of microtubule-binding proteins of the outer kinetochore by Aurora kinases results in weakened interactions with incoming microtubules and thus facilitates the resolution of incorrect attachments resulting in error correction (Welburn et al., 2010; DeLuca et al., 2011). Uncongressed chromosomes close to the poles of the mitotic spindle come in to proximity of Aurora A. Aurora A, like Aurora B, can thus phosphorylate sites in the outer kinetochore implicated in microtubule binding and thus promote release of incorrectly positioned chromosomes from the mitotic spindle (DeLuca et al., 2018; Ye et al., 2015).

Both Aurora kinases also promote chromosome biorientation and spindle bipolar assembly by inhibiting the activity of kinesin motor proteins. MCAK and KIF18B, which regulate the stability of microtubules attached to kinetochores are inhibited by Aurora A– and B–dependent phosphorylation (Andrews et al., 2004; Ems-McClung et al., 2013; McHugh et al., 2019; Tanenbaum

et al., 2011; Zhang et al., 2008). Aurora A also phosphorylates and inhibits the centromere-associated kinesin CENP-E involved in efficient chromosome congression (Kapoor et al., 2006; Kim et al., 2010). A phosphorylation site mutant form of CENP-E cannot be inhibited by Aurora A and thus traps nonbioriented chromosomes at the spindle poles (Kapoor et al., 2006; Kim et al., 2010). Aurora A kinase is therefore thought to promote chromosome congression and error correction through inhibitory mechanisms that prevent trapping of chromosomes at the poles of the mitotic spindle.

This simple model poses a problem, since nonequatorial chromosomes will be released from bound microtubules by either the action of Aurora A or B and thus fail to undergo movement toward the cell equator. How is congression of these nonequatorial chromosomes actively promoted? One possibility is that Aurora kinases, in addition to inhibitory actions, also facilitate chromosome congression by modulating the strength of the polar ejection force generated by chromokinesins.

Chromokinesins are a small subgroup of the kinesin superfamily that have binding sites for both microtubules and DNA (Almeida and Maiato, 2018; Mazumdar and Misteli, 2005). The ability to bind both to chromosomes and microtubules is thought to enable chromokinesins to generate the polar ejection force. This aids chromosome congression to the metaphase plate by pushing the chromosome and chromosome arms away from the poles and toward the cell equator (Rieder and Salmon, 1994). Chromokinesins KID and KIF4A of the kinesin-10 and kinesin-4

[1]Department of Biochemistry, University of Oxford, Oxford, UK;   [2]Sir William Dunn School of Pathology, University of Oxford, Oxford, UK.

Correspondence to Francis A. Barr: francis.barr@bioch.ox.ac.uk.

families, respectively, are the best studied. Initially, KID was thought to be the major kinesin creating the polar ejection force in human cells (Brouhard and Hunt, 2005; Levesque and Compton, 2001). However, subsequent studies showed that effective chromosome congression in cells requires cooperation of KID with KIF4A (Stumpff et al., 2012; Wandke et al., 2012).

KIF4A is particularly interesting in this context, because it is known to be regulated by Aurora B (Bastos et al., 2014; Nunes Bastos et al., 2013). Early in mitosis, KIF4A plays an important role in both chromosome alignment and chromosome condensation (Mazumdar et al., 2004; Samejima et al., 2012; Stumpff et al., 2012; Takahashi et al., 2016; Wandke et al., 2012). In anaphase, KIF4A regulates microtubule dynamics and central spindle length (Kurasawa et al., 2004; Zhu and Jiang, 2005). These different functions are reflected in the different localizations of KIF4A. Throughout mitosis, KIF4A localizes to chromatin, while in anaphase, a pool of KIF4A relocates to the spindle midzone (Mazumdar et al., 2004). The temporally and spatially distinct functions of KIF4A are mediated by specific interaction partners at these different sites: the structural maintenance of chromosomes (SMC) protein complex condensin I on chromatin (Takahashi et al., 2016), and the microtubule-binding and bundling protein regulator of cytokinesis 1 (PRC1) at the central spindle, respectively (Gruneberg et al., 2006; Kurasawa et al., 2004; Zhu and Jiang, 2005).

Condensin complexes localize to the axes of chromosomes in mitosis and are essential for chromatin compaction and chromosome organization (Hirano, 2012). Mammalian cells have two distinct condensin complexes, condensin I and II, which share the SMC subunits SMC2 and SMC4 but differ in the composition of their non-SMC subunits, CAP-D, G, and H (Ono et al., 2003). KIF4A localizes to chromosome arms and interacts specifically with the condensin I complex (Takahashi et al., 2016). In the absence of KIF4A, chromosomes become overcondensed and shortened along their axes (Samejima et al., 2012; Takahashi et al., 2016). In anaphase, a second pool of KIF4A interacts with PRC1, a highly conserved, homodimeric microtubule-binding protein required for central spindle formation (Mollinari et al., 2002, 2005). PRC1 and KIF4A form a minimal self-organizing microtubule bundling system in anaphase (Hannabuss et al., 2019) and are mutually dependent on each other for localization to the anaphase spindle midzone, where they together determine central spindle length and contribute to anaphase chromosome position (Bieling et al., 2010; Hu et al., 2011; Pamula et al., 2019; Subramanian et al., 2010). This is achieved by molecular tagging of the ends of microtubules in the central spindle zone by KIF4A and PRC1 in a microtubule-length–dependent manner (Subramanian et al., 2013). Phosphorylation of KIF4A at T799 close to the motor domain by the mitotic kinase Aurora B stimulates the motor activity of KIF4A and is necessary for the anaphase function of KIF4A (Nunes Bastos et al., 2013).

While it is clear that condensin I and PRC1 contribute to the differential localization of KIF4A to distinct cellular structures, the specific signal in KIF4A that mediates the interaction with PRC1 has not been identified. Furthermore, it is unclear whether activation of the KIF4A motor activity by Aurora kinases is relevant for the function of KIF4A and condensin I in chromosome

condensation and congression. The interaction of KIF4A with condensin I is not directly regulated by Aurora B (Poonperm et al., 2017), eliminating this simple possibility. To understand the function of KIF4A at chromatin, we therefore identified and expanded the motifs in KIF4A required for interaction with condensin I and PRC1, respectively. This enabled us to analyze the different aspects of KIF4A functionality in isolation and test the idea that Aurora kinases contribute to chromosome congression by stimulating the motor activity of the chromosome bound pool of the chromokinesin KIF4A.

## Results

### Isolation of KIF4A mutants disrupted for either condensin I or PRC1 binding

In anaphase, KIF4A is present midway between the segregating chromosomes in a region defined by the anaphase spindle midzone protein PRC1 (Fig. S1 A). In both metaphase and anaphase, a discrete population of KIF4A concentrates on the axes of chromosomes similar to the condensin subunit SMC2 (Fig. S1 B). To understand the distinct functions of KIF4A at the chromosome axis and central spindle, we searched for specific point mutants in KIF4A that would disrupt the interactions with either condensin I or PRC1. Alignment of the C-terminal tail of metazoan KIF4A reveals the presence of a series of conserved motifs (Fig. 1 A). The conserved C-terminal region of KIF4A containing the cysteine-rich domain (CRD) has previously been implicated in chromatin binding (Wu and Chen, 2008) and interaction with PRC1 (Hu et al., 2011). In addition, the IXE motif of KIF4A, amino acid sequence SPIEEE, has been implicated in binding to the phosphatase PP2A-B56 (Hertz et al., 2016). Between these two regions are three absolutely conserved sequences. Two diphenylalanine clusters, FF1154 and FF1220, flank a positively charged lysine/arginine basic patch. The second of these motifs, FF1220, is needed for condensin I binding (Takahashi et al., 2016). Because diphenylalanine associated with a charge cluster has been implicated in other protein–protein interactions (Kaiser et al., 2005; Loewen et al., 2003), we focused on this region in our search for the PRC1-binding motif.

To test the role of these conserved features, they were either deleted or mutated to alanine and the corresponding KIF4A complexes isolated by immunoprecipitation from HeLa cells in anaphase, when both the PRC1 and condensin I interactions can be detected (Fig. 1 B). This revealed that the KIF4A FF1154AA mutant fails to interact with PRC1 but retains its ability to bind condensin I detected using CAP-G and SMC2 (Fig. 1 C). Conversely, the FF1220AA and basic patch mutants failed to bind condensin I but still interacted with PRC1 (Fig. 1 C). As expected, deletion of the entire KIF4A tail region or a double FF1154AA and 1220AA point mutant disrupted binding to both PRC1 and condensin I (Fig. S1 C). Deletion of the CRD reduced, but did not abolish, binding to condensin I and had no effect on the PRC1 interaction.

### Condensin I and PRC1-binding mutants of KIF4A show specific localization defects

We next explored the roles of the CRD, basic patch, FF1154, and FF1220 motifs on KIF4A localization in metaphase and anaphase.

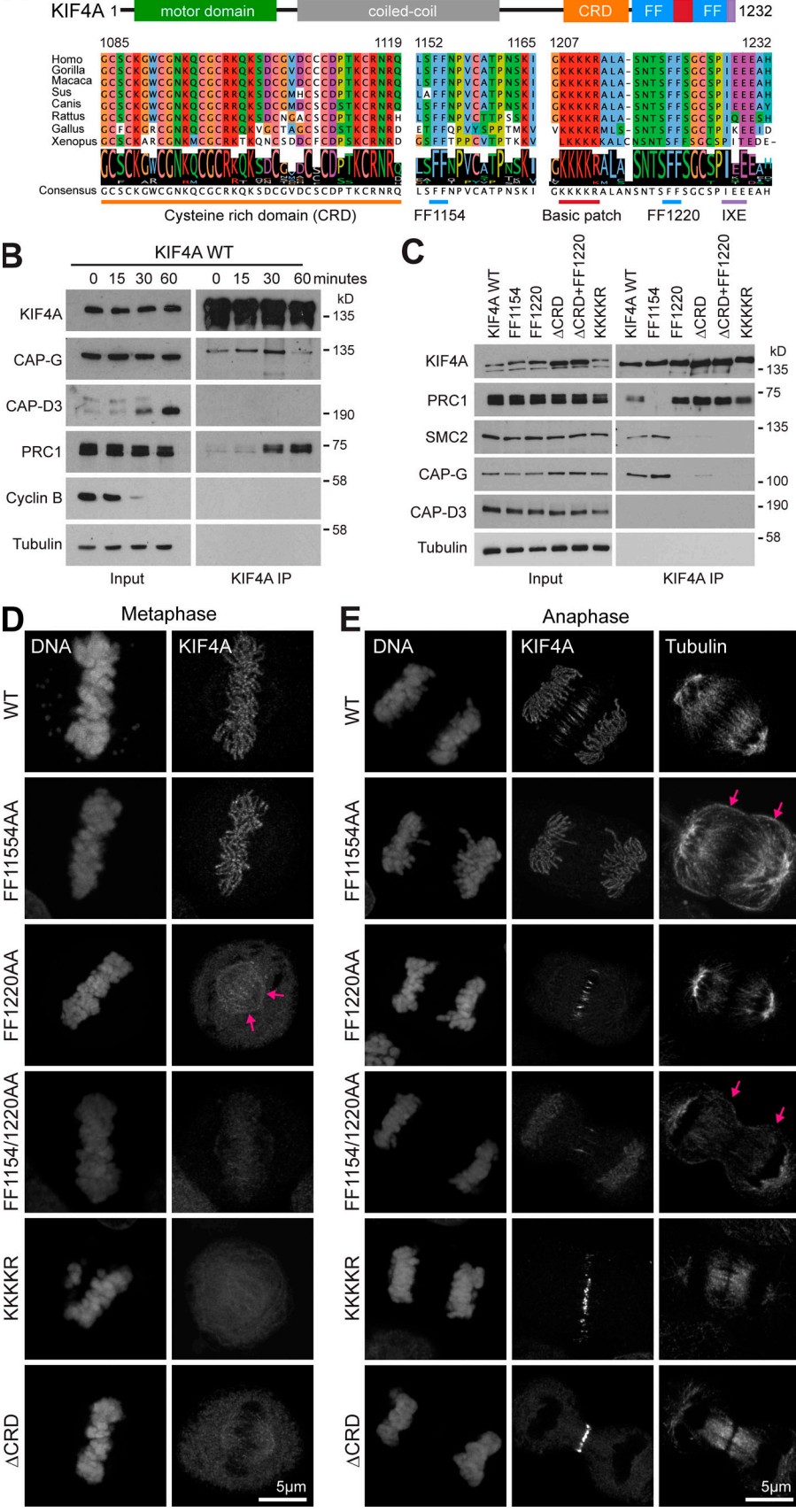

**Figure 1. Two conserved FF motifs in the C-terminal tail of KIF4A support interaction with either condensin I or PRC1. (A)** Sequence alignment of the KIF4A C-terminal tail from a group of metazoans reveals a high degree of conservation. **(B)** Endogenous KIF4A was immunoprecipitated from HeLa cells exiting mitosis. A 0-min time point was taken 25 min after nocodazole washout. Synchronous progression of mitotic cells into anaphase was triggered with 2 μM MPS1 inhibitor (AZ3146) to override the spindle checkpoint. The temporal association of KIF4A to condensin I and PRC1 was followed over time by Western blot. **(C)** Immunoprecipitation (IP) of WT FLAG-KIF4A (WT), deletion mutants, and alanine point mutants from anaphase HeLa cells. **(D and E)** Superresolution images of FlpIn T-REx eGFP-KIF4A HeLa cells induced with doxycycline to express KIF4A WT and point mutants in metaphase (D) and anaphase (E). Arrows mark the weak metaphase spindle staining for KIF4A FF1220AA and the cortical spread of microtubules in the FF1154AA PRC1-binding mutant and FF1154/1220AA double mutant in anaphase.

For this purpose, HeLa Flp-In T-REx cell lines expressing inducible copies of WT or mutant GFP-KIF4A were generated. In comparison to WT GFP-KIF4A, the FF1154AA PRC1-binding–defective mutant failed to localize to the anaphase spindle but showed otherwise normal targeting to chromosome axes in both metaphase and anaphase (Fig. 1, D and E). Compared with the control condition, anaphase microtubules were spread around the cell cortex (Fig. 1 E, pink arrows). The ΔCRD, basic patch, and FF1220 mutants all failed to target to the chromosome axes in either metaphase or anaphase but were present on the central spindle in anaphase (Fig. 1, D and E). Interestingly, the FF1220AA mutant showed weak metaphase spindle staining (Fig. 1 D, pink arrows). The FF1154AA/1220AA double mutant, which failed to bind either PRC1 or condensin I (Fig. S1 C), was defective for chromatin and central spindle targeting (Fig. 1, D and E). Like the FF1154AA single mutant, anaphase microtubules were spread around the cell cortex in the FF1154AA/1220AA double mutant (Fig. 1 E, pink arrows). Together, these data support the view that targeting to the chromosome axis and central spindle are independent events mediated through different pathways (Takahashi et al., 2016). This conclusion is also consistent with the analysis of KIF4A complexes in cells exiting mitosis (Fig. 1, B and C), where the FF1154AA and FF1220AA, ΔCRD, or basic patch mutants were defective for either PRC1 or condensin I binding, respectively.

Because temporal control of KIF4A is crucial for its function, the effects of the FF1154AA and FF1220 mutations on KIF4A complexes during mitosis and mitotic exit were followed (Video 1, Video 2, Video 3, and Video 4). This confirmed that the FF1154AA mutation is defective for PRC1 interaction but shows condensin I binding in metaphase and anaphase (Fig. 2, A–C). In contrast, the FF1220AA mutant was defective for condensin I binding at all time points but still displayed anaphase binding to PRC1 (Fig. 2, A–C). The diphenylalanine clusters at FF1154 and FF1220 thus define motifs required for interaction of KIF4A with PRC1 and condensin I, respectively. In addition to FF1220, the upstream CRD and basic patch regions are required for efficient condensin I binding but appear to play no role in PRC1 binding (Fig. 1 C). Time-lapse imaging demonstrated that these mutants showed the localization defects matching these binding defects. KIF4A FF1154, which cannot bind PRC1, was present only on chromatin, whereas the ΔCRD and FF1220 mutants localize only to the anaphase spindle (Fig. 2 D). A double FF1154AA and 1220AA mutant was localized to the nucleus in G2 but failed to show any distinct localization in either metaphase or anaphase (Fig. 2 D). These mutants therefore enable us to investigate the specific functions of KIF4A–condensin I and KIF4A–PRC1 complexes.

## PRC1-binding mutants of KIF4A are defective for central spindle length control

Two key hallmarks are associated with of loss of KIF4A and PRC1 function in anaphase. First, the integrity of the central spindle is lost and microtubule bundles become disorganized and fail to form an ordered centered array, and second, chromosomes show enhanced separation and movement in anaphase (Nunes Bastos et al., 2013; Pamula et al., 2019). KIF4A is crucial for anaphase spindle length control and the formation of organized bundles of anaphase spindle microtubules (Fig. S2 A). When it is depleted, PRC1, components of the MKLP1-centralspindlin complex, and cytokinesis factors such as ECT2 spread out along microtubules rather than exhibit a focused localization to antiparallel microtubule overlap regions at the cell equator (Fig. S2 B). To test the specific role of the KIF4A–PRC1 interaction in anaphase spindle elongation, WT and mutant GFP-KIF4A were expressed in HeLa cells depleted of endogenous KIF4A with siRNA directed to the 3′ UTR of the KIF4A mRNA. Anaphase chromosome separation was increased in cells expressing the FF1154AA/mutant compared with WT KIF4A or the condensin I–binding–defective FF1220AA and ΔCRD KIF4A mutants (Fig. S1 D). In addition, the PRC1 signal was spread over a broader region of the anaphase B spindle in cells expressing the FF1154AA mutant (Figs. 3 A and S1 E). These findings support the view that the FF1154AA mutant is defective for the anaphase functions of KIF4A, while the FF1220AA and ΔCRD KIF4A mutants are not.

To further investigate the role of the KIF4A–PRC1 interaction for PRC1 function in spindle organization, we performed time-lapse imaging of HeLa Flp-In T-Rex cells depleted of endogenous KIF4A and coexpressing GFP-PRC1 and either mCherry-KIF4A WT or the PRC1-binding–defective FF1154AA mutant. PRC1 was recruited to the anaphase spindle with similar timing in KIF4A WT or FF1154AA-expressing cells but was spread over a broader region of the anaphase B spindle in the mutant (Fig. 3, B and C). Furthermore, chromosome separation was increased during anaphase in the FF1154A mutant cells (Fig. 3 D). Measurements of PRC1 distribution at the cell equator show that the overall size of the cell does not change (bright-field images), but in FF1154AA-expressing cells, PRC1 distribution spreads out both along the long axis of the cell and toward the cell cortex in anaphase B compared with KIF4A WT (Fig. 3, B, E, and F). This may explain why microtubules spread out toward the cell cortex and lose their clustered organization in the FF1154AA mutant condition (Fig. 1 E), similar to the effect of KIF4A knockdown (Fig. S2 A). A further consequence is that cell furrowing initiates earlier in the KIF4A FF1154AA mutant than the WT condition (Fig. 3, B, E, and F). Similar effects were seen in cells that do not express GFP-PRC1 (Fig. S2, C–E), confirming this effect is due to expression of the FF1154AA mutant.

Analysis of the 3D distribution of PRC1 and the MKLP1–centralspindlin complex was then performed in anaphase cells before and during furrowing. Timings were derived according to the extent of chromosome separation using time-lapse imaging data (Fig. 3, B–F). Consistent with other work (Pamula et al., 2019), we observed that PRC1 localized to bundles centered in the middle of the cell away from the cortex early in anaphase before furrowing (Fig. 4 A, WT 6–8 min), and this organization persists once furrowing commences (Fig. 4 A, WT 14–16 min). In the FF1154AA mutant, PRC1 spreads out toward the cell cortex, and the central bundles are lost (Fig. 4 A, FF1154AA 6–8 min). This becomes

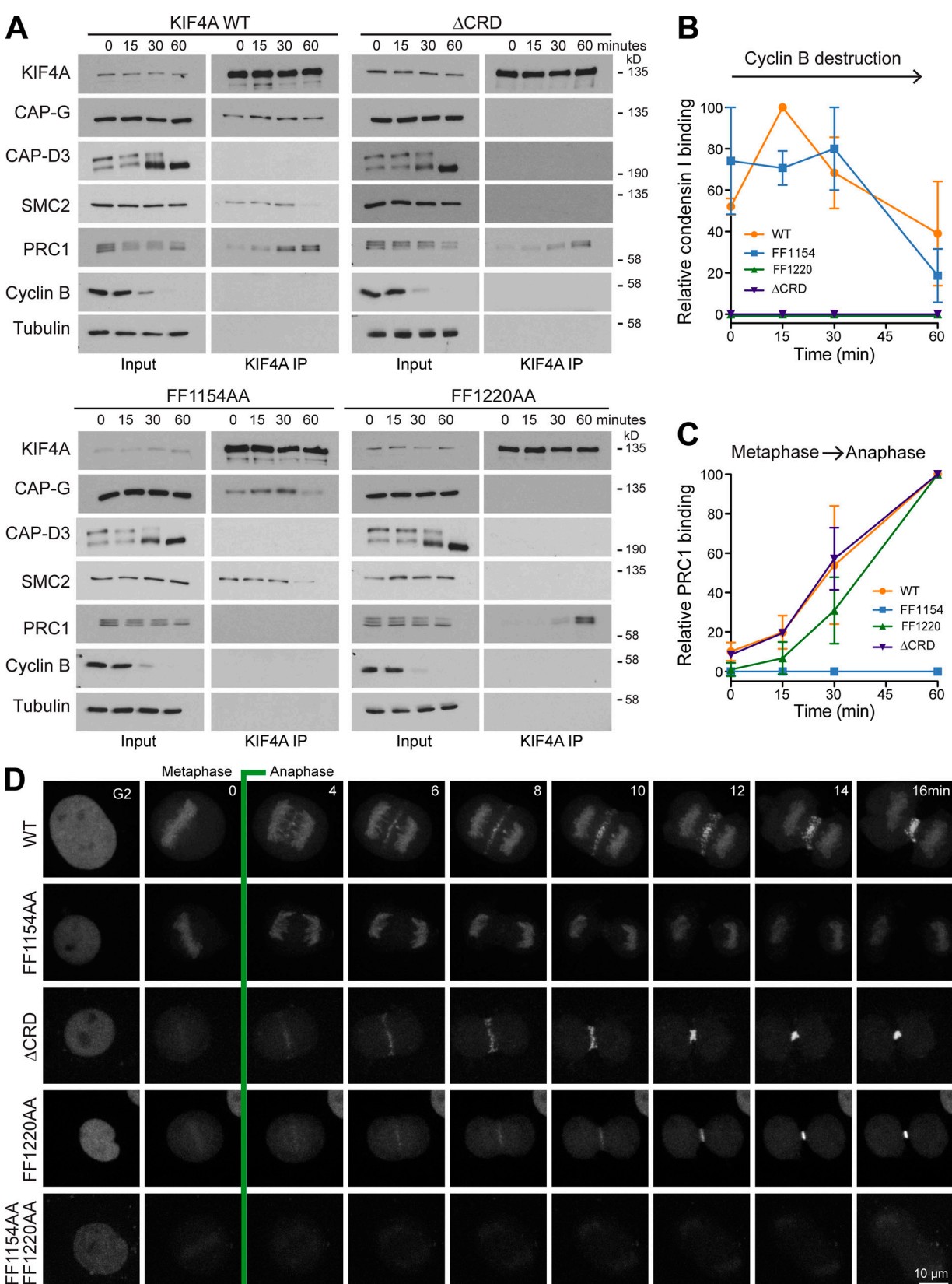

Figure 2. **Temporal regulation of FF1220 condensin I– and FF1154 PRC1-binding mutants of KIF4A. (A)** FLAG-tagged KIF4A WT and mutant complexes were isolated from cells progressing from metaphase into anaphase (KIF4A IP). The temporal association of KIF4A to condensin I and II and PRC1 was followed over time using Western blot. **(B and C)** Relative KIF4A association to condensin I (B) and PRC1 (C) was measured as a function of time and the mean plotted in the line graphs. Error bars indicate the SD for *n* = 2. **(D)** HeLa cells depleted of endogenous KIF4A and expressing fluorescent protein tagged KIF4A WT, FF1154AA, ΔCRD, or FF1220AA were imaged undergoing mitosis. Representative maximum projections are shown.

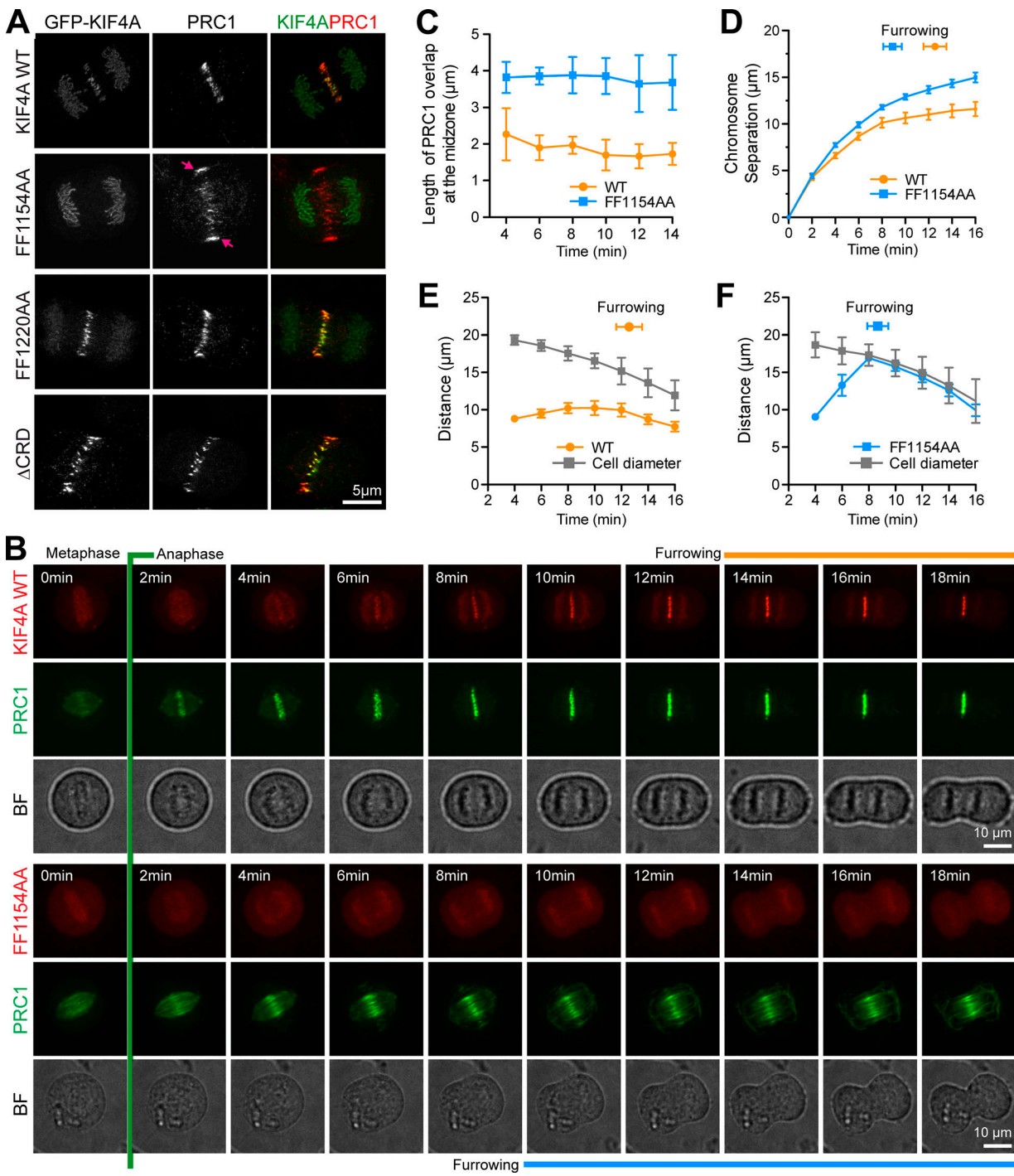

Figure 3. **Defective anaphase spindle structure in the FF1154AA PRC1-binding mutant of KIF4A. (A)** Superresolution images of WT KIF4A and mutants, showing anaphase localization of KIF4A and PRC1. Arrows mark the spread of PRC1 signal at the central spindle. **(B)** HeLa Flp-In T-Rex cells depleted of endogenous KIF4A induced to express GFP-PRC1 and cotransfected with either mCherry-KIF4A WT or the FF1154AA mutant were imaged every 2 min while entering anaphase. Brightfield (BF) reference images are also shown. **(C–F)** Graphs shows the length of the PRC1 overlap at the midzone (C) and chromosome separation as a function of time for the WT and the FF1154AA mutant (D). The start of furrowing is marked. Plots in E and F measure the distribution of PRC1 and the diameter of the cell during mitotic exit for the WT and the FF1154AA mutant. For PRC1 overlap in C, the mean and SD are plotted; in all other cases, the mean and SEM are plotted (WT $n = 7$, FF1154AA $n = 10$).

most obvious when viewed along the long axis of the spindle using a 90° rotation of the image data (Fig. 4 A, 90° rotation right panels). Once cytokinesis commences later in anaphase, PRC1 is concentrated into a defined ring close to the cell cortex

in KIF4A F1154A cells, whereas the defined microtubule bundles seen in the WT remain spatially separated from the cell cortex (Fig. 4 B, FF1154AA 14–16 min). Similar effects were seen with the centralspindlin protein MKLP1 (Fig. 4 B),

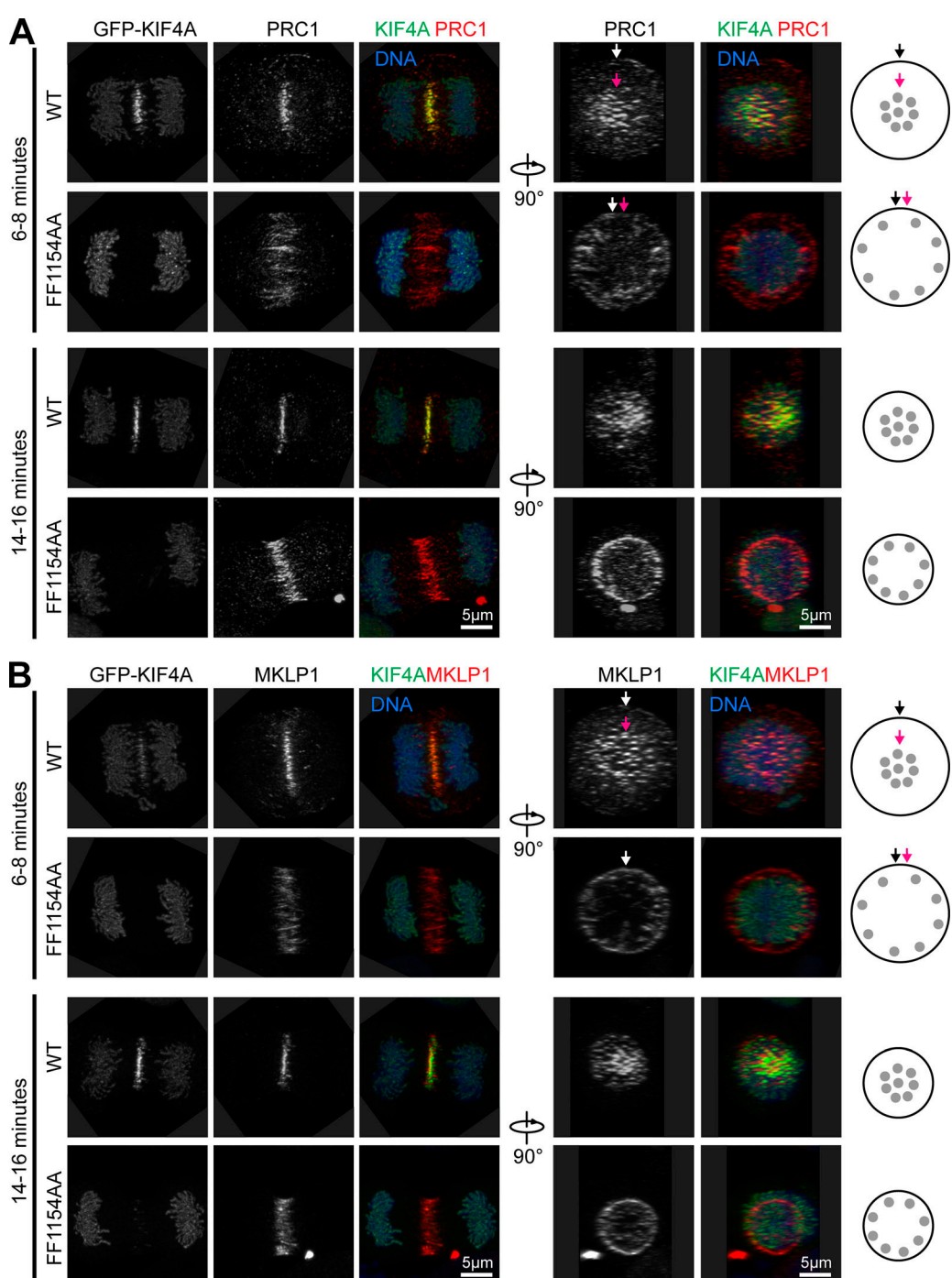

Figure 4. **KIF4A FF1154 fails to support radial compaction of the anaphase central spindle. (A)** Superresolution images of KIF4A WT and the FF1154AA mutant, showing anaphase localization of KIF4A and PRC1 before furrowing (6–8 min) and during furrowing (14–16 min). Stacks were acquired to capture the total volume on the cell and the radial distribution of PRC1 at the cell equator is showed in the 90° rotation. **(B)** Superresolution images of WT KIF4A and the FF1154AA mutant showing the distribution of KIF4A and MKLP1 in anaphase before furrowing (6–8 min) and during furrowing (14–16 min). Stacks were acquired to capture the total volume on the cell and the radial distribution of MKLP1 at the cell equator is showed in the 90° rotation. The white arrows mark the edge of the cell, and the pink arrows mark the bundles of PRC1 and MKLP1. Cartoons on the right depict PRC1 and MKLP1 distribution in WT and FF1154AA cells.

suggesting that KIF4A–PRC1 acts upstream of this complex in mediating anaphase spindle organization.

Taken together, these data support the view that KIF4A is specifically required for regulation of anaphase spindle elongation through its interaction with PRC1 rather than acting as a global regulator of microtubule dynamics. These findings also highlight the indirect role played by KIF4A and PRC1 in the regulation of other factors at the central spindle, including the centralspindlin complex, required for the correct spatial and temporal regulation of cytokinesis.

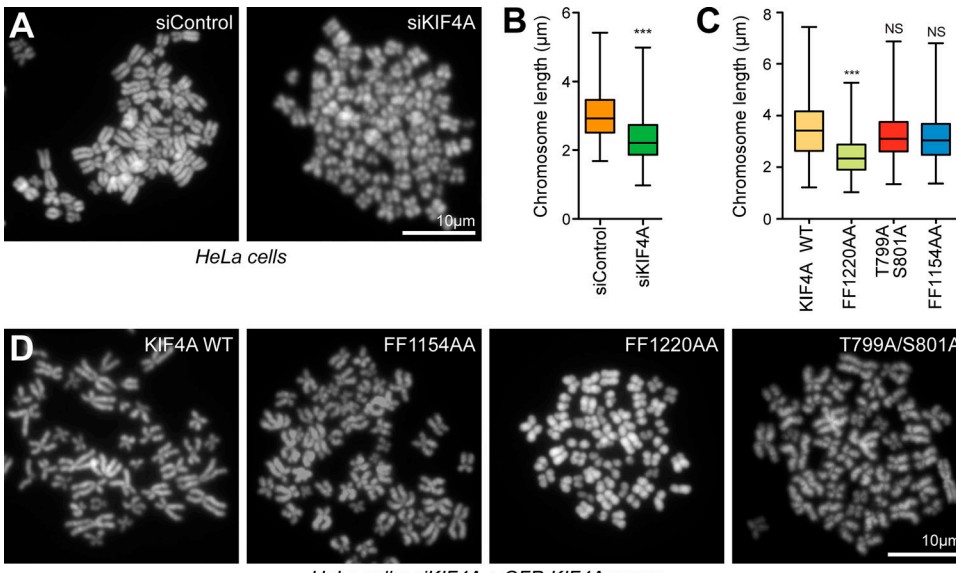

Figure 5. **Chromosome condensation requires the KIF4A–condensin I interaction, but not Aurora-dependent phosphorylation or PRC1 binding. (A)** Chromosome spreads of HeLa cells transfected with control or siKIF4A. **(B)** Box and whiskers plots of the chromosome length measured in FIJI (siControl n = 368, siKIF4A n = 425). The mean is indicated, and whiskers show minimum and maximum values. An unpaired t test with Welch's correction and 99% confidence intervals was performed (***, P < 0.001). **(C)** Box and whiskers plots for chromosome length measured in FIJI from spreads of HeLa cells treated with siKIF4A and transfected with KIF4A WT and the indicted mutants (WT KIF4A n = 515, FF1220AA n = 492, T799A/801A = 653, and FF1154AA n = 660). A nonparametric Kruskal-Wallis test to KIF4A was performed (P < 0.0001), and conditions were compared post analysis by Dunn's test (***, P < 0.001). **(D)** Representative images of chromosome spreads analyzed in C are shown.

## Condensin I–binding mutants of KIF4A are defective for chromosome morphology

The role of KIF4A in chromosome morphology was then investigated. As previously reported (Samejima et al., 2012), depletion of KIF4A results in significantly increased axial compaction of chromosomes in chromosome spread preparations (Fig. 5, A and B). This effect was seen in both HeLa and hTERT-RPE1 (human telomerase immortalized retinal pigmented epithelial cells; Fig. S3, A and B) and thus appears to reflect a general function for KIF4A in chromosome structure. The dependency of this function on the interaction with condensin I, PRC1, and requirement for Aurora-dependent phosphorylation was then examined. For this purpose, WT and mutant GFP-KIF4A were expressed in HeLa cells depleted of endogenous KIF4A with siRNA directed to the 3′ UTR of the KIF4A mRNA (Fig. S3 C). In this assay, WT KIF4A and the PRC1-binding–defective FF1154AA mutant rescued the axial compaction of chromosomes (Fig. 5, C and D). However, chromosomes in the condensin I–binding–defective FF1220AA mutant showed significantly increased axial compaction (Fig. 5, C and D), in agreement with previous work (Takahashi et al., 2016). The KIF4A–condensin I interaction is therefore important for normal chromosome structure. Importantly, the T799A/S801A phosphorylation mutant deficient for Aurora-regulated kinesin motor activity rescued axial compaction in this assay (Fig. 5, C and D). This agrees with previous work showing that Aurora kinases do not regulate the KIF4A–condensin I interaction (Poonperm et al., 2017) and also suggests that the function of KIF4A in chromosome structure does not require Aurora-regulated kinesin motor activity. KIF4A therefore has discrete roles at chromosomes and the anaphase central spindle dependent on its specific binding partners at these sites.

## The condensin-dependent pool of KIF4A promotes chromosome alignment

KIF4A has also been implicated in chromosome movement and the generation of the polar ejection force promoting congression of nonequatorial chromosomes to the metaphase plate (Stumpff et al., 2012; Wandke et al., 2012). This role overlaps with that of the second chromokinesin KID, albeit the precise role of each kinesin remains unclear. However, since KIF4A and KID localize to different regions of the chromosome, differences in their functions are expected. KIF4A is present along the chromosome axis, and KID shows a diffuse distribution across the chromosome (Fig. 6 A). To follow the role of KIF4A in chromosome congression, CRISPR/Cas9–engineered cells expressing a GFP-tagged version of the condensin subunit SMC2 from the endogenous locus were used, since this enables individual chromosome arms to be clearly visualized. Depletion of both KIF4A and KID gave rise to chromosome alignment defects and delayed progression through mitosis (Fig. 6 B, arrows; Fig. S4, A and B; and Video 5). Time-lapse imaging showed that most chromosomes were aligned at the metaphase plate with normal kinetics after 30 min, but a small population remained at the spindle poles and showed delayed congression into the metaphase plate (Fig. S4 A, arrows). Furthermore, the movement of the chromosomes arms into the plate was delayed and only completed after ~50–60 min of mitosis. The ability of WT or mutant KIF4A forms to rescue these defects was then tested (Video 6, Video 7, and Video 8). As expected, the PRC1-binding–defective FF1154AA mutant supported chromosome congression and was indistinguishable from WT KIF4A (Fig. S4 C). By contrast, the condensin I–binding–defective FF1220AA mutant failed to rescue chromosome congression and showed

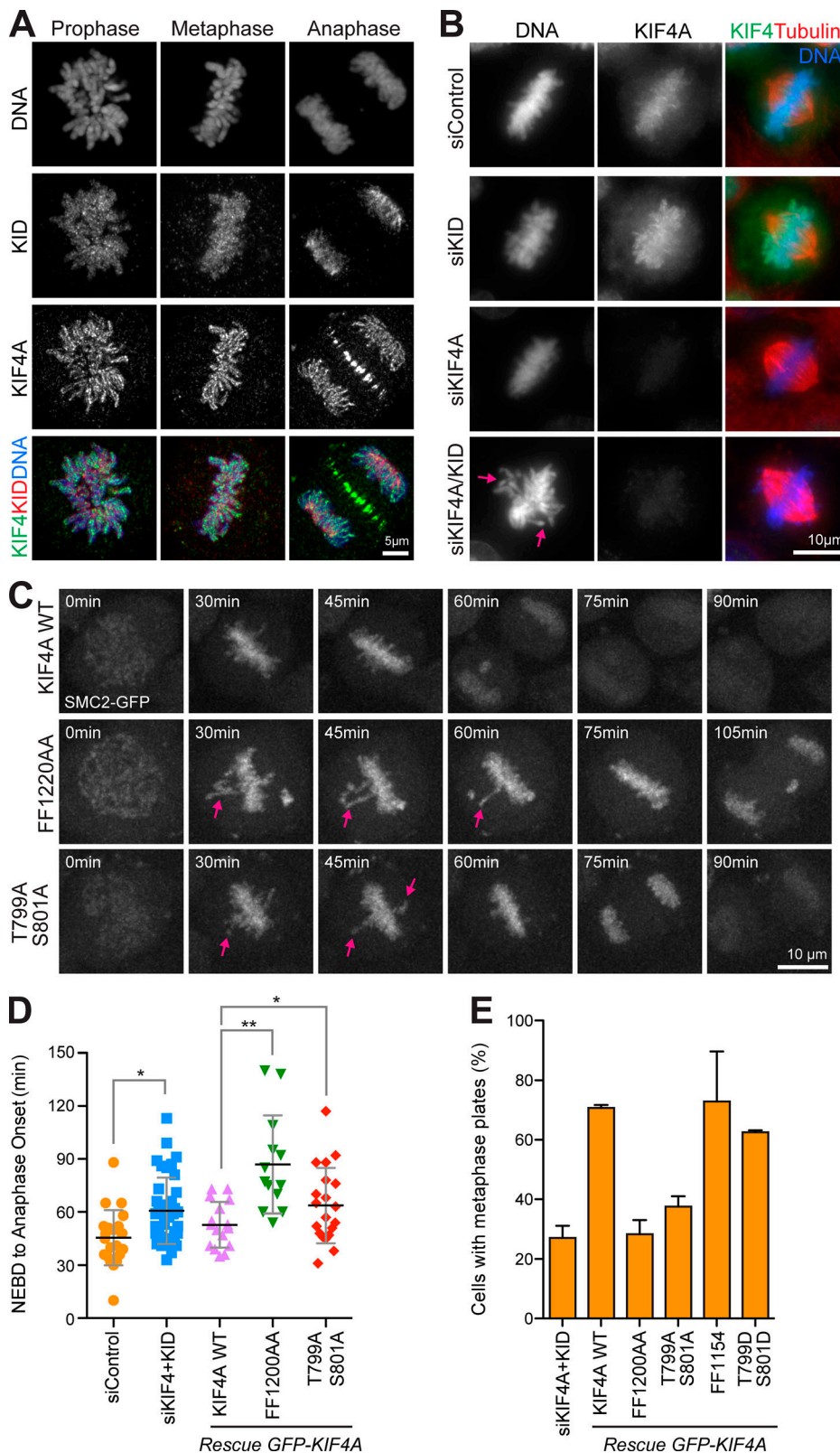

Figure 6. **KIF4A, but not the T799/S801 phosphorylation site mutant, promotes prometaphase chromosome congression. (A)** Localization of KIF4A and KID in mitosis. CRISPR-tagged EGFP-KIF4A cells were fixed and stained for endogenous KID. Images were acquired with an Airyscan system. **(B)** HeLa cells were transfected for 48 h before thymidine synchronisation with control, siKIF4A, siKID, and siKIF4A + KID. After thymidine release and fixation, cells were stained for endogenous KIF4A. Arrows mark chromosomes and chromosome arms showing delayed congression to the metaphase plate. **(C)** mScarlet-KIF4A WT, FF1220AA, or T799A/S801A were expressed in a CRISPR-edited SMC2-EGFP cell line codepleted for KIF4A and KID. Cells were imaged undergoing mitosis, and representative maximum projections of the SMC2-EGFP signal are shown. Arrows mark chromosomes and chromosome arms showing delayed

congression to the metaphase plate. **(D)** The plot shows the mean time and individual data points from nuclear envelope breakdown (NEBD) to anaphase onset in SMC2-EGFP cells transfected with either siControl or siKIF4A + KID and rescued with KIF4A WT and mutants (siControl n = 22, siKIF4A+KID n = 59, rescue KIF4A WT n = 16, FF1220AA n = 14, and T799A/S801A n = 20). Error bars indicate the SD. A nonparametric Kruskal-Wallis test (P < 0.0001) with a 95% confidence interval was performed, and conditions were compared after analysis by Dunn's test (**, P < 0.001; *, P < 0.1). **(E)** HeLa cells codepleted of KIF4A and KID and transfected with GFP-KIF4A constructs were fixed and stained for DNA and tubulin. The bar graph displays the extent of chromosome alignment for KIF4A and KID codepleted cells and for cells expressing GFP-KIF4A WT and mutants (siKIF4A+KID n = 103, rescue KIF4A WT n = 98, FF1154AA n = 73, FF1220AA n = 56, T799A/S801A n = 60, and T799A/S801A n = 66). Error bars indicate the SD (for three independent experiments).

significantly delayed progression into anaphase (Fig. 6, C–E; and Fig. S4 C). To examine the requirement for Aurora phosphorylation, phosphomimetic and phosphorylation-site–defective mutants were compared. Compaction of chromosome arms into the metaphase plates was delayed until 50–60 min of mitosis in cells expressing the T799A/S801A Aurora phosphorylation site mutant (Figs. 6 C and S4 C). Cells expressing the T799A/S801A mutant showed delayed progression into anaphase (Fig. 6 D) and formed fewer metaphase plates (Figs. 6 E and S4 C). Consistent with the idea that Aurora phosphorylation promotes KIF4A function, a phosphomimetic T799D/S801D mutant was able to rescue chromosome congression to nearly the same level as KIF4A WT (Figs. 6 E and S4 C).

Chromosome congression therefore requires both recruitment of KIF4A to the axes of chromosomes by condensin I and Aurora regulation of kinesin motor activity. Notably, this is different to the function in chromosome condensation, which is supported by the T799A/S801A phosphorylation mutant and therefore does not require Aurora-regulated kinesin motor activity.

### KIF4A phosphorylation by Aurora A at unaligned chromosomes

Aurora A and B create defined nonoverlapping phosphorylation gradients within the spindle and at the spindle poles and centromeres, respectively (Fig. 7 A). To assess the contributions Aurora A and B make to KIF4A regulation in chromosome alignment, their role in T799/S801 phosphorylation in prometaphase was examined using specific antibodies. Consistent with a function for Aurora-dependent phosphorylation of KIF4A both in early and late mitosis, KIF4A phosphorylation, detected with T799 phosphoantibodies (pT799), rises as cells enter mitosis with a similar profile to the T288-activating phosphorylation on Aurora A (pT288) and the S10 phosphorylation of histone H3 (pS10) created by Aurora B (Fig. S5 A). This rise in pT799 is slightly delayed when compared with cyclin-dependent kinase mediated phosphorylation events detected using PRC1 pT481, and furthermore persists longer into anaphase (Fig. S5 B, 12- to 13-h time points). In cells with unaligned chromosomes, KIF4A pT799 is detected at regions overlapping with both Aurora A and B at the spindle poles and centromeres, respectively (Fig. 7 B). These signals are lost when KIF4A is depleted, leaving only a nonspecific centrosome signal at each spindle pole (Fig. 7 B). KIF4A has been previously shown to be a target of Aurora B (Nunes Bastos et al., 2013), and we therefore tested if it can also be phosphorylated by Aurora A. In vitro kinase assays showed that recombinant KIF4A can be phosphorylated by Aurora A and that this activity is inhibited by the specific Aurora A inhibitor MLN8537 (Fig. 7 C). We then investigated the contribution of Aurora A and Aurora B to

KIF4A phosphorylation at the different regions of the mitotic spindle in prometaphase. For this purpose, cells were treated with a CENP-E inhibitor to accumulate unaligned chromosomes close to the spindle poles.

In control cells, unaligned chromosomes close to the Aurora A polar region or undergoing alignment at the equator show punctate pT799 signals (Fig. 7 D, arrow A). A strong punctate pT799 signal overlapping with the centromere marker CENP-A is seen on aligned chromosomes at the spindle equator (Fig. 7 E, arrow B). When Aurora A was inhibited, the pT799 signal on the polar, nonequatorial chromosomes was lost within 10 min (Fig. 7, D and E, Aurora A inhibition, A), while the signal on aligned equatorial chromosomes was retained (Fig. 7, D and E, Aurora A inhibition, B). Inhibition of Aurora B resulted in reduction of the equatorial signal for pT799 (Fig. 7, D and E, Aurora B inhibition, B). Measurement of the pT799 signal in control and Aurora A and B inhibition conditions showed these were producible changes (Fig. 7 F). Western blot shows that, under these conditions in living cells, the two inhibitors display the expected specificity for Aurora A (pT288) and Aurora B (pS10), respectively (Fig. 7 G). In summary, we conclude that the pT799 signal reflects the overlap between distributions of KIF4A, Aurora A, and Aurora B. Notably, Aurora A can phosphorylate KIF4A on unaligned chromosomes close to the spindle poles in prometaphase.

### Aurora A–dependent chromosome congression requires KIF4A

To directly test for a specific role of Aurora A in KIF4A-dependent chromosome alignment, we used an assay for bipolar spindle formation from monastrol-induced monopolar spindles. This assay has been previously shown to recapitulate chromosome congression to the metaphase plate and Aurora B–dependent error correction, and we reasoned it may also be Aurora A dependent (Kapoor et al., 2006). We modified the assay to include transient inhibition of either Aurora A or Aurora B after the monastrol washout step (Fig. 8 A). Following Aurora A inhibition, a population of chromosomes remained at the spindle poles in >50% of cells depleted of KIF4A, but not in the control cells (Fig. 8, B and C). Fewer than 20% of cells depleted of KID or treated with an Aurora B inhibitor showed unaligned chromosomes (Fig. 8, B and C). Chromosome alignment from the Aurora A–inhibited state therefore requires KIF4A, but not KID.

Further developing this approach, we then asked which properties of KIF4A are required for Aurora A–dependent chromosome congression. Metaphase plates formed in ~40% of cells following Aurora A inhibition, and depletion of endogenous KIF4A and expression of GFP-KIF4A WT increased this to >75% (Fig. 8, D and E). Condensin I binding FF1220AA and T799A/S801A Aurora-phosphorylation mutants aligned

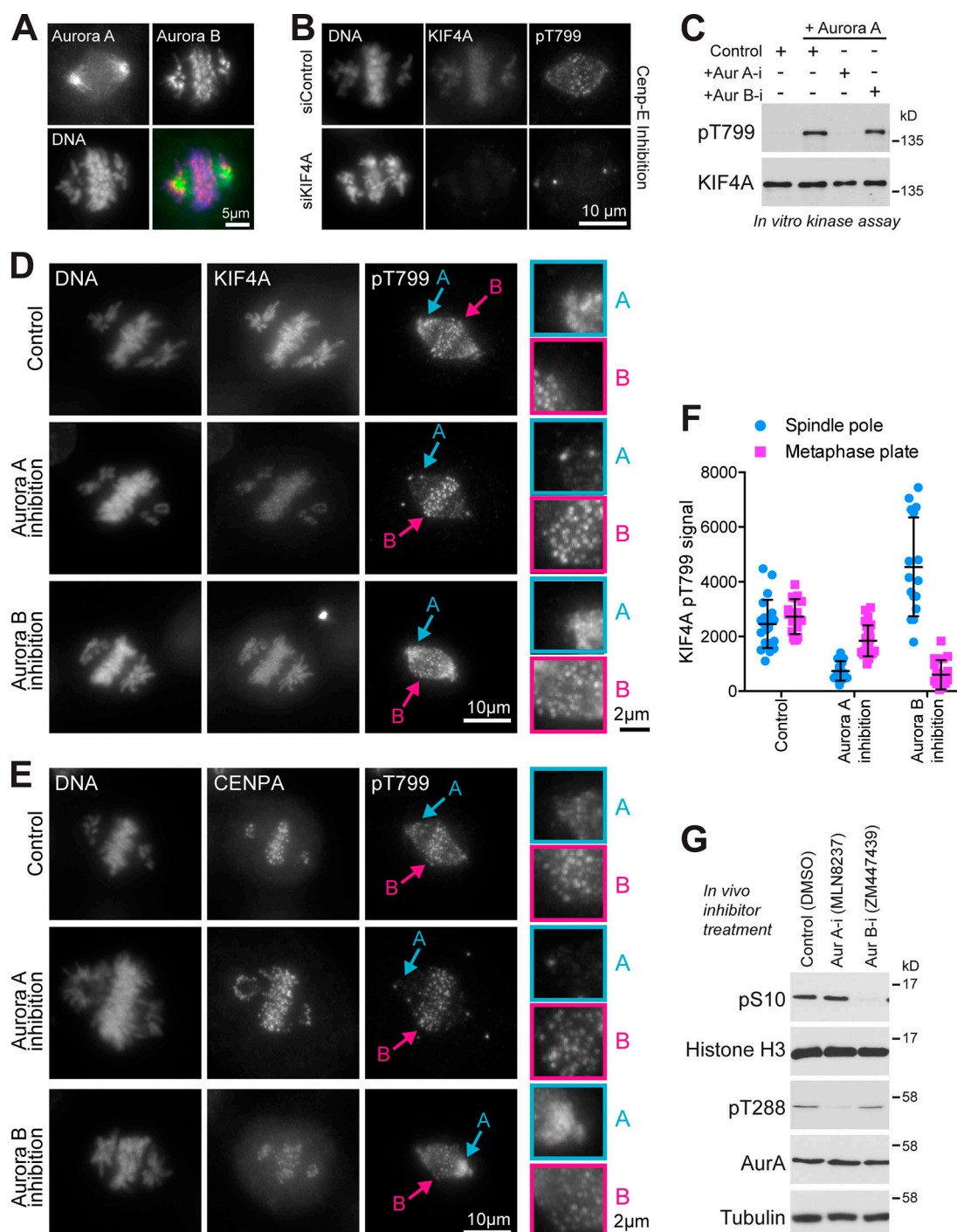

Figure 7. **Aurora A phosphorylates KIF4A on off-axis chromosomes. (A)** At 7 h after thymidine release, HeLa cells were treated with 300 nM CENP-E inhibitor (GSK923295) for 3 h to accumulate chromosomes at the spindle poles. The cells were then stained for DNA, Aurora A, and Aurora B. **(B)** CRISPR-edited EGFP-KIF4A cells transfected with either siControl or siKIF4A for 48 h and treated with CENP-E inhibitor for 1 h were stained for KIF4A pT799 and DNA. **(C)** Purified KIF4A was treated with Aurora A in the presence or absence of Aurora A or B inhibitors. The samples were Western blotted for total KIF4A and pT799. **(D and E)** KIF4A pT799 staining compared with KIF4A (D) and CENP-A (E) following 10 min treatment with DMSO (control), Aurora A inhibitor (Aur A-i; 0.5 µM MLN8237), or Aurora B inhibitor (Aur B-i; 0.2 µM ZM447439). Arrows A and B mark the phosphorylation signals of KIF4A on chromosomes in the Aurora A and B regions of the spindle, respectively. The enlargements on the right show the pT799 signal in the same regions. **(F)** The graph shows the intensity of pT799 signal in control (*n* = 10), Aurora A inhibited (*n* = 11), and Aurora B inhibited (*n* = 10) cells at the spindle poles and on the metaphase plate. Error bars and SD are shown. **(G)** Western blots show the specificity of the Aurora A and B inhibitors in cells under these conditions.

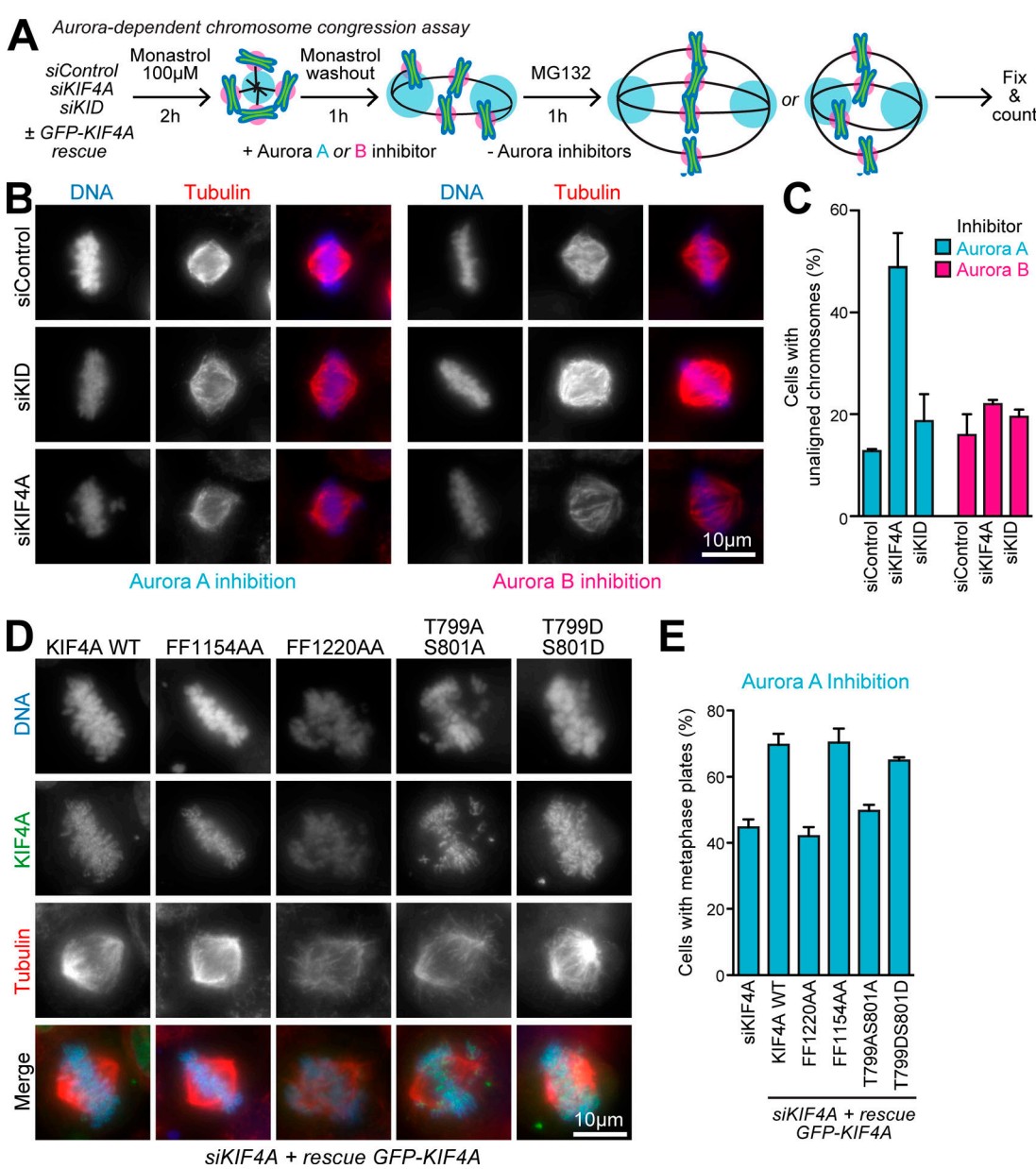

Figure 8. **Aurora A regulation of KIF4A is required for efficient chromosome congression. (A)** A schematic of the chromosome congression assay. **(B)** HeLa cells transfected with siControl, siKIF4A, or siKID for 48 h were treated with 10 µm monastrol for 2 h. Monastrol was removed and replaced with inhibitors for either Aurora A (0.5 µM MLN8237) or Aurora B (0.2 µM ZM447439) for 1 h. Cells were then washed into fresh medium and after a further 1 h fixed and then stained for tubulin and DNA. **(C)** The percentage of cells with unaligned chromosomes is plotted in the graph (Aurora A inhibition: siControl $n = 77$, siKIF4A $n = 149$, and siKID $n = 74$; Aurora B inhibition: siControl $n = 109$, siKIF4A $n = 101$, and siKID $n = 132$ from two independent experiments). Error bars indicate the SD. **(D)** HeLa cells transfected with siKIF4A for 24 h and transfected with GFP-tagged KIF4A rescue constructs for a further 24 h were treated with 10 µm monastrol for 2 h. Monastrol was removed by washing in fresh growth medium and replaced with medium containing Aurora A inhibitor (0.5 µM MLN8237) for 1 h. Cells were then washed into fresh medium and after a further 1 h, fixed, and then stained for tubulin and DNA. **(E)** The percentage of cells with aligned chromosomes is plotted in the graph (siKIF4A $n = 93$, KIF4A WT $n = 96$, FF1154AA $n = 96$, FF1220AA $n = 145$, T799A/S801A $n = 82$, and T799D/S801D $n = 60$ from two independent experiments). Error bars indicate the SD.

all chromosomes to metaphase plates in only 40–45% of cases (Fig. 8, D and E). The FF1154AA PRC1-binding mutant not expected to show defects in prometaphase behaved like KIF4A WT in this assay (Fig. 8, D and E). Finally, consistent with the idea that Aurora phosphorylation promotes KIF4A function, the phosphomimetic T799D/S801D mutant was able to rescue alignment of chromosomes to the metaphase plate to nearly the same level as KIF4A WT (Fig. 8, D and E).

## Discussion

This work provides mechanistic insight into how Aurora-kinase gradients within the mitotic spindle promote chromosome congression. We find that Aurora A–dependent regulation of a pool of KIF4A recruited to the chromosome axes through interaction with condensin I is important for the congression of chromosomes into a compact metaphase plate. This role is different to the function of KIF4A in chromosome

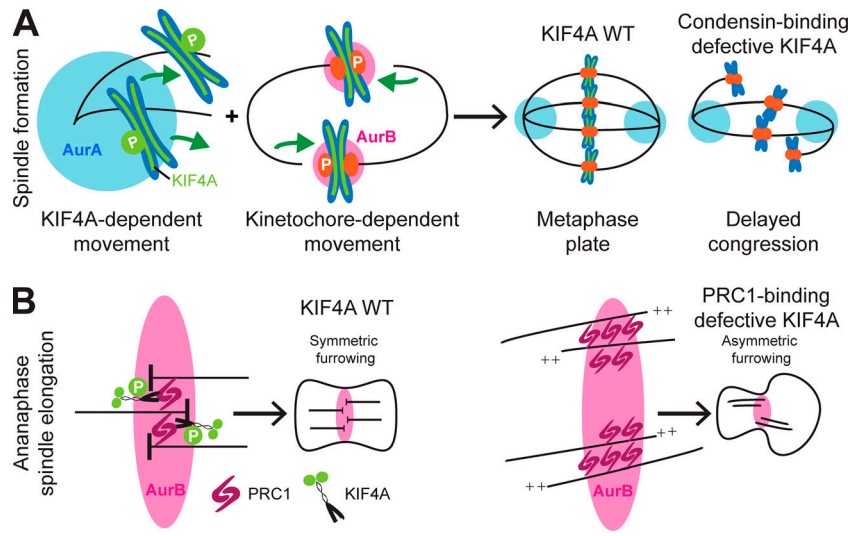

**Figure 9. Models for KIF4A function chromosome alignment and anaphase spindle formation. (A)** The model summarizes Aurora-regulated kinetochore-dependent and -independent pathways contributing to chromosome congression. At kinetochores, Aurora B destabilizes kinetochore–microtubule attachments for nonbioriented chromosomes. Aurora A phosphorylates the condensin I–bound pool of KIF4A at polar chromosomes, promoting KIF4A motor activity and chromosome congression. **(B)** In anaphase, PRC1 and KIF4A control microtubule growth and the organization of the anaphase spindle, ensuring symmetric furrowing. The KIF4A mutant FF1154AA shows this interaction is essential for regulated microtubule growth and organized cytokinesis.

structure, which also requires interaction with condensin I (Samejima et al., 2012; Takahashi et al., 2016) but as we show here does not require the phosphorylation-stimulated kinesin motor activity. Aurora kinases therefore play a crucial role in activating KIF4A function during chromosome movements and thus complement the kinetochore and CENP-E–dependent capture and biorientation pathway (Fig. 9 A). Previous work has explained how Aurora A and B uncouple chromosomes from microtubules by phosphorylation of outer kinetochore proteins and the kinesin motor CENP-E (DeLuca et al., 2011; DeLuca et al., 2018; Kim et al., 2010; Welburn et al., 2010). We now propose that Aurora A phosphorylates and activates KIF4A, thus promoting the kinetochore-independent congression of chromosomes and chromosome arms from the cell poles into the metaphase plate (Fig. 9 A). Without this pathway, nonequatorial chromosomes would be released from bound microtubules by either the action of Aurora A or B and thus fail to undergo movement toward the cell equator. This failure is seen in cells expressing the condensin I and Aurora-phosphorylation mutants of KIF4A. We therefore conclude that Aurora kinases facilitate chromosome congression by modulating the strength of the KIF4A-dependent component of the polar ejection force. Our data also demonstrate that the interaction of KIF4A is necessary for proper regulation of chromosome structure by condensin I and that this effect is independent of Aurora-regulated motor activity or the interaction with PRC1 required for microtubule organization at the anaphase central spindle (Fig. 9 B). However, this leaves open the question of how KIF4A acts. One simple possibility is that the dimeric C-terminal tail of KIF4A, which has two copies of FF1220, would act to link two condensin I complexes. Further work will be necessary to test this idea. The specific loss-of-function mutations that we describe here, targeting different aspects of KIF4A function, will greatly facilitate this work as well as studies of the role of KIF4A and PRC1 at the anaphase spindle.

As already discussed, we provide important new insights into the regulation of the polar ejection force by Aurora A.

Nonetheless, a number of questions relating to this process remain to be addressed. KIF4A phosphorylation is lost within 10 min of Aurora A inhibition, and thus, there must be a counteracting phosphatase to explain this finding. Previous reports show that PP2A-B56 interacts with KIF4A at the central spindle and dephosphorylates T799 in anaphase (Bastos et al., 2014; Hertz et al., 2016). However, KIF4A does not show colocalization with PP2A-B56 along the chromosome axes early in mitosis. Additionally, the binding site mapping data, summarized in Fig. 1 A, indicate that the condensin I–binding site is immediately adjacent to the PP2A-B56–binding motif and that binding of these two large multisubunit complexes is thus likely to be mutually exclusive. This suggests that there is another phosphatase or pool of PP2A-B56 that acts on chromatin-bound KIF4A. One possibility is that this could be PP1, as in the case of CENP-E (Kim et al., 2010). Further questions relate to the regulation of KID and its precise mechanism of localization to chromatin in prophase, prometaphase, and anaphase. Answering these will be important to fully understand high-fidelity capture and segregation of chromosomes by the mitotic spindle.

## Materials and methods

### Reagents and antibodies

General laboratory chemicals and reagents were obtained from Sigma-Aldrich and Thermo Fisher Scientific unless specifically indicated. Inhibitors for Aurora A (MLN8237; Cambridge Bioscience), Aurora B, (ZM447439; Tocris Bioscience), MPS1 (AZ3146; Tocris Bioscience), KIF11 (Monastrol; Tocris Bioscience) and CENP-E (GSK923295; Selleckchem) were dissolved in DMSO to make 10-mM stocks. MG132 (20-mM stock; Calbiochem) was dissolved in DMSO. Thymidine (100-mM stock; Sigma-Aldrich) and doxycycline (1-mM stock; Invivogen) were dissolved in water. A 1-mg/ml stock of the DNA dye Hoechst 33258 (Sigma-Aldrich) was dissolved in water.

Commercial antibodies were used to detect PRC1 pT481 (rabbit monoclonal 2189–1; Epitomics), cyclin B1 (mouse monoclonal 05–373; EMD Millipore), tubulin (mouse monoclonal

T6199; Sigma-Aldrich), and FLAG epitope tag (rabbit polyclonal F7425; Sigma-Aldrich). SMC2 (rabbit polyclonal ab10399; Abcam), NCAP-G (rabbit polyclonal A300-602A; Bethyl), NCAP-D3 (rabbit polyclonal ab70349; Abcam), KID/KIF22 (rabbit polyclonal ab222187; Abcam), Aurora A (rabbit polyclonal 4718S; Cell Signaling), Aurora B (mouse monoclonal AIM1; Becton Dickinson), Aurora A pT288 (rabbit polyclonal 3079S; Cell Signaling), H3 (rabbit polyclonal 4499S; Cell Signaling), histone H3 pS10 (mouse monoclonal 6G3; Cell Signaling), and CENP-A antibody (mouse monoclonal ab13939; Abcam). Noncommercial sheep antibodies to GFP, mCherry, ECT2, KIF23 (MKLP1), and KIF4A and rabbit antibodies to PRC1 and KIF4A pT799 have been reported previously (Cundell et al., 2013; Nunes Bastos et al., 2013). Secondary donkey antibodies against mouse, rabbit, or sheep and labeled with Alexa Fluor 488, Alexa Fluor 555, Alexa Fluor 647, Cy5, or HRP were purchased from Molecular Probes and Jackson ImmunoResearch Laboratories, Inc., respectively. The RFP booster Atto 594-conjugate was purchased from Chromotek. Affinity purified primary and HRP-coupled secondary antibodies were used at 1 µg/ml final concentration. For Western blotting, proteins were separated by SDS-PAGE and transferred to nitrocellulose using a Trans-blot Turbo system (Bio-Rad). All Western blots were revealed using ECL (GE Healthcare). Protein concentrations were measured by Bradford assay using Protein Assay Dye Reagent Concentrate (Bio-Rad).

### Molecular biology
All DNA primers were obtained from Invitrogen. Human KIF4A was amplified using Pfu hot start turbo polymerase (Agilent Technologies). Mammalian expression constructs were made in pcDNA5/FRT/TO (Invitrogen) modified to encode EGFP, mScarlet, or FLAG tags. Mutagenesis was performed using the Quick-Change method (Agilent Technologies). CRISPR/Cas9 guide RNA cassettes targeting sequences in human KIF4A (5′-TCTCAGCAC CGTCTCTGCGAGGG-3′ and SMC2 (5′-AACTTCCACATGTGCTCC TT-3′) were constructed in pSpCAS9(BB) (Addgene) modified to remove the puromycin resistance marker. Human KIF4A siRNA 5′-CAGGTCCAGACTACTACTC-3′ against the 3′ UTR was obtained from QIAgen, and an optimized siRNA pool for human KIF22 (KID) was obtained from Dharmacon (L-004962-00-0005). Target sequences for KIF22 are 5′-GCUGGUCGCUGUCGGCUAA-3′, 5′-GAGAGCGGAUGGUGCUAAU-3′, 5′-UUGAGAGGCUUAAGA CGAA-3′, and 5′-CCAAGGAGGUGAUCAAUCG-3′. Control siRNA duplexes were directed toward firefly luciferase 5′-CGUACGCGG AAUACUUCGA-3′.

### Cell culture procedures
HeLa cells and HEK293T were cultured in DMEM with 1% [vol/vol] GlutaMAX (Life Technologies) containing 10% [vol/vol] bovine calf serum at 37°C and 5% $CO_2$. RPE1 cells were cultured in DME/F-12 (Sigma-Aldrich) supplemented with 1% [vol/vol] GlutaMAX (Invitrogen) and containing 10% [vol/vol] bovine calf serum at 37°C and 5% $CO_2$. Mirus LT1 (Mirus Bio) was used to transfect all cell lines. siRNA transfection was performed with Oligofectamine (Invitrogen) and TransIT-X2 (Mirus Bio) for HeLa and RPE, respectively. HeLa cell lines with single integrated copies of PRC1, KIF4A WT,

and the mutants described in the figures were created using the T-REx doxycycline-inducible Flp-In system (Invitrogen). HeLa cell lines with GFP inserted in frame at the N-terminus of KIF4A or the C-terminus of SMC2 in the endogenous loci were constructed using CRISPR/Cas9 following a published protocol with some modifications (Alfonso-Perez et al., 2019; Stewart-Ornstein and Lahav, 2016). In brief, homology recombination cassettes containing the desired knock-in DNA with homology regions of 600–750 bp flanking the target locus were cotransfected with versions of pSpCAS9(BB) containing the KIF4A or SMC2 guide RNAs. The knock-in sequences harbor the EGFP fluorescent protein, a glycine-serine rich flexible linker (GSSS repeated four times), a P2A ribosome skipping sequence, and a resistance marker (puromycin or blasticidin). Antibiotic resistant clones were selected with puromycin to generate the KIF4A cell line and blasticidin for the SMC2 cell line, and successful modification was confirmed by blotting.

### Isolation of KIF4A complexes and in vitro kinase assays
For each time course, four 15-cm dishes of HeLa cells, one dish per time point, were transfected with FLAG-KIF4A constructs. After 24 h, cells were arrested with 0.1 µM nocodazole for 18 h. Mitotic cells were collected by shake-off, pooling together dishes transfected with the same construct. Nocodazole was removed by washing three times with PBS and then twice with growth medium prewarmed to 37°C and equilibrated with $CO_2$. Cells were then resuspended in 4.5 ml growth medium and left in the incubator at 37°C for 25 min to allow the reconstitution of the mitotic spindle. After harvesting a 0-min time point by washing 1 ml of cells in ice cold PBS, 2 µM AZ3146 (MPS1 inhibitor) was added to the remaining cell suspension and further samples taken at 15, 30, and 60 min. Cells were lysed on ice with 1 ml of lysis buffer (20 mM Tris-HCl, pH 7.4, 150 mM NaCl, 1% [vol/vol] Igepal CA-630, 0.1% [wt/vol] sodium deoxycholate, 100 nM okadaic acid, 40 mM β-glycerophosphate, 10 mM NaF, 0.3 mM $Na_3VO_4$, protease inhibitor cocktail [Sigma-Aldrich], phosphatase inhibitor cocktail [Sigma-Aldrich]) supplemented with 40 U benzonase nuclease (E1014; Sigma-Aldrich) for 3 mg of total lysate and 2 mM of $MgCl_2$. FLAG-KIF4A complexes were isolated by 2-h incubation at 4°C with 15 µl anti-FLAG M2-agarose beads (Sigma-Aldrich). Beads were washed twice with 1 ml lysis buffer and twice with 1 ml wash buffer (20 mM Tris-HCl, pH 7.4, 150 mM NaCl, 0.1% [vol/vol] Igepal CA-630, 40 mM β-glycerophosphate, 10 mM NaF, and 0.3 mM $Na_3VO_4$) and resuspended in 50 µl 3x Laemmli buffer for Western blotting. KIF4A in vitro kinase assays were performed using recombinant Aurora A (Nunes Bastos et al., 2013). In brief, FLAG-KIF4A purified from HEK 293T cell was incubated for 30 min at 37°C in the absence or presence of recombinant Aurora A with DMSO (control), 1 µM MLN8237 (Aurora A inhibitor), or 1 µM ZM447439 (Aurora B inhibitor). Samples were analyzed by blotting for KIF4A and pT799.

### Fixed-cell imaging and superresolution microscopy
For fixed-cell imaging, cells were grown on No 1.5 glass coverslips (AGL46R16-15; Agar Scientific), washed once with PBS, and then fixed in –20°C methanol for 5 min. Coverslips were washed

three times with PBS and then stained. Primary and secondary antibody staining was performed in PBS for 60 min at room temperature. Coverslips were mounted in Moviol4-88 mounting medium (EMD Millipore). Images were acquired using a 60× NA 1.35 oil immersion objective) on an upright microscope (BX61; Olympus) with filter sets for DAPI, GFP/Alexa Fluor 488, Alexa Fluor 555, Alexa Fluor 568, and Alexa Fluor 647 (Chroma Technology) an 2,048 × 2,048–pixel complementary metal oxide semiconductor camera (PrimΣ; Photometrics), and MetaMorph 7.5 imaging software (Molecular Devices). Illumination was provided by an LED light source (pE300; CoolLED Illumination Systems). Image stacks of 10–17 planes with a spacing of 0.6 µm through the cell volume were maximum intensity projected in MetaMorph.

For high-resolution imaging, doxycycline-inducible Flp-In T-REx WT or mutant EGFP-KIF4A HeLa cells were seeded on 22 × 22 × 0.17–mm coverslips and transfected with siRNA against the 3′ UTR of endogenous KIF4A. After 48 h, cells were synchronized by addition of thymidine to a final concentration of 2.5 mM for 20 h. Cells were then washed three times in fresh growth medium, and expression of KIF4A was then induced with 1 µM doxycycline for 10 h. Fixation was performed by immersing the cells on coverslips in –20°C methanol for 5 min and then washing three times in 2 ml room temperature PBS before staining with antibodies for tubulin and PRC1 as described in the figures. When the SMC2-GFP cell line was used, cells were transfected with mCherry-KIF4A constructs and the signal was enhanced by using an RFP-booster Atto 594 conjugate. Tubulin and PRC1 were stained with antibodies. DNA was detected using DNA dye Hoechst 33258 diluted to 1 µg/ml. In all cases, coverslips were mounted on slides using a 50-µl drop of Vectashield (Thermo Fisher Scientific). Images were acquired using an inverted Zeiss LSM880 microscope fitted with an Airyscan detector and a Plan-Apochromat 63×/NA 1.4 oil lens. The 488-nm argon and 561-nm diode lasers in combination with a dual band emission filter (band pass 420–480 nm and 495–550 nm) were used either to excite GFP and mScarlet or detect staining of tubulin and PRC1. The DNA signal was detected using a solid-state laser. Sequential excitation of each wavelength was switched per Z-stack. Airyscan processing was performed using the ZEN system software. Z-stack maximum intensity projection was performed using FIJI (Schindelin et al., 2012) to produce TIFF images for the figures. Images were then cropped in Adobe Photoshop and placed into Illustrator to produce the figures. To visualize cells in 3D, an average of 100 stacks (0.14 µm between Z-stacks) were acquired. The processed images were then opened in FIJI and a 3D projection with interpolation was performed. The 3D rendered images were saved both as ".avi" movie files and sequences of TIFF images for use in the figures.

### Live-cell microscopy of KIF4A-PRC1 and SMC2

For live-cell imaging, cells were plated in 35-mm dishes with a 14-mm No. 1.5 thickness coverglass window (MatTek Corporation) and then treated as described in the figure legends. To follow PRC1 localization in mitosis, Flp-In T-REx HeLa cells expressing inducible GFP-PRC1 were used. The cells were transfected with siKIF4A 3′ UTR for 24 h to deplete endogenous KIF4A and then transfected with mCherry-KIF4A WT or

FF1154AA mutant for a further 18 h. Expression of PRC1 was induced with doxycycline for 5–6 h prior the start of imaging. To follow chromosome alignment, the CRISPR-edited GFP-SMC2 cells were used. The dishes were placed in a 37°C and 5% $CO_2$ environment chamber (Tokai Hit) on an inverted microscope (IX81; Olympus) with a 60× 1.42 NA oil-immersion objective coupled to an Ultraview Vox spinning disk confocal system running Volocity 6 (PerkinElmer). Images were captured with an electron multiplying charge coupled device camera (C9100-13; Hamamatsu Photonics). Exposure time was 100 ms 4% 488-nm laser power for GFP and 6% 561-nm lasers for mScarlet. Typically, 15 planes, 0.6 µm apart, were imaged every minute. Spindle length measurements were performed on the original data with Volocity 6 software. Images for figures in TIFF format and movies were created in FIJI using maximum intensity projection of the fluorescent channels. Images were then placed into Adobe Illustrator CS5. Measurements of spindle length, cell diameter, and PRC1 distribution were performed in Volocity. The diameter of the cell was measured according to the bright field.

### Chromosome spreads

HeLa S3 and hTERT-RPE1 cells were seeded at low confluency ($3 × 10^6$ cells per 15-cm dish) and then transfected with siControl or siKIF4A after 24 h. After a further 24 h, cells were either left untransfected or transfected with EGFP-KIF4A constructs for 40 h. At this stage, dishes were at 60–70% confluency, and the cells were arrested with nocodazole 0.1 µm/ml for 6 h. Mitotic cells were then collected by shake off, and a small aliquot was saved to check depletion and overexpression levels. Cells were resuspended in 0.075 mM KCl prewarmed at 37°C to allow cell swelling. After 20–30 min of incubation at 37°C, cells were pelleted at 1,000 rpm for 5 min and resuspended in the leftover volume of KCl. Fixative was prepared by mixing 3 vol methanol and 1 vol acetic acid on ice. Fixation was performed adding gently 500 µl ice-cold fixative while vortexing at slow speed. Before performing the spread, super frost slides (AGL4341; Agar Scientific) were placed in cold water. Cells were resuspended in cold fixative and dropped onto a slide tilted at 45°C, washed gently with fixative solution, and dried on a wet paper towel placed on top of a water bath set at 80°C. Coverslips were then mounted of the slides using Mowiol 4–88 containing 1 µg/ml Hoechst 33258. Chromosomes were imaged using a 100× NA 1.4 immersion oil objective taking a snapshot of a single plane. Images were then opened and analyzed in FIJI, and chromosomes length was measured manually.

### Chromosome congression assays

To analyze the Aurora dependence of chromosome congression in a synchronized system, we modified and extended a published assay used to study the dynamics of chromosome alignment (Kapoor et al., 2006). Briefly, asynchronous cultures of HeLa cells were treated with 10 µM monastrol for 2 h to trap cells entering mitosis with monopolar spindles. To promote spindle formation, the cells were then washed three times in PBS and three times in prewarmed growth medium. Aurora A (0.5 µM MLN8537) or Aurora B inhibitor (0.2 µM ZM447439) and 20 µM of the proteasome inhibitor MG132 were then added. After 1 h of inhibitor treatment,

cells were washed three times in PBS and twice in prewarmed growth media and then released for a further 1 h in growth media containing 20 µM MG132 to prevent cyclin B destruction and hence anaphase onset. Coverslips were then fixed in methanol and stained for DNA, tubulin, and KIF4A. Chromosome alignment to the metaphase plate was counted and plotted in the figures.

## Statistical analysis

Statistical significance was analyzed using a Kruskal-Wallis test, Dunn's test, or unpaired $t$ test with Welch's correction, as indicated in the figure legends. For the $t$ test, data distribution was assumed to be normal, but this was not formally tested. Statistical significance was determined using GraphPad Prism software version 5.

## Online supplemental material

Fig. S1 shows KIF4A localization in mitosis and provides support to Figs. 1 and 3. Fig. S2 extends the analysis of the anaphase spindle and cytokinesis in Fig. 3. Fig. S3 shows chromosomes spreads for hTERT-RPE1 cells and extends Fig. 5. Data on chromosome alignment are shown in Fig. S4 to supplement Fig. 6. Fig. S5 shows analysis of KIF4A phosphorylation by Western blot to supplement Fig. 7. Image data showing KIF4A localisation as a function of time from Fig. 3 D is complemented by videos for KIF4A WT (Video 1), and for three mutants: FF1154AA (Video 2), ΔCRD (Video 3), and FF1220AA (Video 4). Video 5 shows chromosome movement in control and siKIF4/KID cells and supports Fig. S3 A. Videos showing chromosome movement in cells expressing KIF4A WT (Video 6) and for two mutants FF1220AA (Video 7) and T799A/S801A (Video 8) supplement Fig. 6 C.

## Acknowledgments

We thank members of the Barr and Gruneberg laboratories for their advice during the work and comments on the manuscript.

This work was supported by the Royal Society Newton Fellowship (grant NF161529 to E. Poser) and Cancer Research UK (program grant C20079/A15940 to F.A. Barr). R. Caous was supported by a Medical Research Council Senior Non-Clinical Research fellowship awarded to U. Gruneberg (MR/K006703/1).

The authors declare no competing financial interests.

Author contributions: Conceptualization, F.A. Barr; Investigation, E. Poser and R. Caous; Funding Acquisition, F.A. Barr and U. Gruneberg; Supervision, F.A. Barr and U. Gruneberg; Writing – Original Draft, F.A. Barr, U. Gruneberg, and E. Poser. Writing – Review & Editing, E. Poser, R. Caous, U. Gruneberg, and F.A. Barr.

Submitted: 26 May 2019

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
