## [Peer Review File · The Journal of Cell Biology]

Aurora A promotes chromosome congression by activating the condensin-dependent pool of KIF4A

Elena Poser, Renaud Caous, Ulrike Gruneberg, and Francis Barr

Corresponding Author(s): Francis Barr, University of Oxford

Review Timeline:

Submission Date:	2019-05-26
Editorial Decision:	2019-07-01
Revision Received:	2019-08-08
Editorial Decision:	2019-09-10
Revision Received:	2019-10-10
Editorial Decision:	2019-10-24
Revision Received:	2019-10-28

Monitoring Editor: Arshad Desai

Scientific Editor: Tim Spencer

Transaction Report:

DOI: <https://doi.org/10.1083/jcb.201905194>

July 1, 2019

Re: JCB manuscript #201905194

Prof. Francis A Barr
University of Oxford
Department of Biochemistry South Parks Road
Oxford OX1 3QU
United Kingdom

Dear Prof. Barr,

Thank you for submitting your manuscript entitled "Aurora A promotes chromosome congression by activating the condensin-dependent pool of KIF4A". Your manuscript has been assessed by two expert reviewers, whose comments are appended below. You will see that both reviewers express potential interest in this work; however, the significant concerns raised by them preclude publication of the current version of the manuscript in JCB.

Both reviewers find your analysis of the two different pools of KIF4A important but they each raise a number of issues related to the experimental analysis and presentation of the work. These issues will need to be addressed in a thorough revision and will be subjected to re-review. We feel that each of the reviewers' comments are on-point and specific and, thus, we hope that you will be able to address all of them in full.

Please let us know if you are able to address the reviewer concerns and wish to submit a revised manuscript to JCB. Note that additional experimental data will be needed to satisfactorily address the concerns of the reviewers. Our typical timeframe for revisions is three to four months; if submitted within this timeframe, novelty will not be reassessed. We would be open to resubmission at a later date; however, please note that priority and novelty would be reassessed.

If you choose to revise and resubmit your manuscript, please also attend to the following editorial points. Please direct any editorial questions to the journal office.

GENERAL GUIDELINES:

Text limits: Character count is < 40,000, not including spaces. Count includes title page, abstract, introduction, results, discussion, acknowledgments, and figure legends. Count does not include materials and methods, references, tables, or supplemental legends.

Figures: Your manuscript may have up to 10 main text figures. To avoid delays in production, figures must be prepared according to the policies outlined in our Instructions to Authors, under Data Presentation, <http://jcb.rupress.org/site/misc/ifora.xhtml>. All figures in accepted manuscripts will be screened prior to publication.

*****IMPORTANT:** It is JCB policy that if requested, original data images must be made available. Failure to provide original images upon request will result in unavoidable delays in publication. Please ensure that you have access to all original microscopy and blot data images before submitting your revision.*******

Supplemental information: There are strict limits on the allowable amount of supplemental data. Your manuscript may have up to 5 supplemental figures. Up to 10 supplemental videos or flash animations are allowed. A summary of all supplemental material should appear at the end of the Materials and methods section.

If you choose to resubmit, please include a cover letter addressing the reviewers' comments point by point. Please also highlight all changes in the text of the manuscript.

Regardless of how you choose to proceed, we hope that the comments below will prove constructive as your work progresses. You can contact the journal office with any questions, cellbio@rockefeller.edu or call (212) 327-8588.

Thank you for thinking of JCB as an appropriate place to publish your work.

Sincerely,

Arshad Desai, PhD
Monitoring Editor
JCB

Tim Spencer, PhD
Deputy Editor
Journal of Cell Biology

Reviewer #1 (Comments to the Authors (Required)):

In this paper, the authors examine the distinct roles of Kif4a in mitosis. Kif4a associates with condensin to target to chromosomes and control chromosome packaging, and with PRC1 to target to the midzone and define the midzone region to measure central spindle length. The authors generate Kif4a mutants that associate with either partners to distinguish their function in these two processes driven by different interaction partners. They identify the motif responsible for the interaction with PRC1 (FF1154), while the other has previously been reported (FF1220). They then test the function of the mutants in chromosome morphology and condensation and in central spindle length. The role of Kif4a in these processes is known from previous studies (mainly RNAi). However, these site-directed mutants are much better tools to probe the specific function of Kif4a and provide the basis for elegant experiments, as shown here. The FF1220 seem to not interact very efficiently with PRC1 however so this motif may be important for interaction with condensing and PRC1.

The second part of the paper about Aurora -regulation of Kif4a is less extensive in the paper. The authors show the non-phosphorylatable mutant on sites S799/T801 (previously identified, Nunes Basto et al 2013) rescues the chromosome compaction defect (from Kif4a phenotype), but does not rescue the chromosome alignment. These findings would indicate that Aurora kinases control Kif4a-mediated chromosome congression but not chromosome packaging. The authors here don't discuss the regulation of Kif4a at midzone by aurora kinases (previously published). Overall the paper and data are good, interesting and solid. However there seem to be an imbalance with a large

emphasis on Aurora regulation of kinesin-4 in the introduction and discussion of the paper, but a modest contribution of these experimental results, possibly a little incomplete on the topic of regulation. Consolidating figure 1-4 into 2/3 figures would rebalance the emphasis of the paper on Aurora regulation of Kif4a, especially a couple of figures or panels don't show anything new (like Kif4a localization).

Major points:

1. The non-phosphorylatable Kif4a mutant rescues the depletion phenotype for compaction, similar to WT Kif4a (Fig 5). However using a phosphoantibody, they show Kif4a on chromosomes is phosphorylated (Fig 7). What happens to the SE mutant? Does it also rescue? Similarly, the authors show S799A/T801A is defective in chromosome alignment. What about the S799E/T801E? Is this one rescuing the phenotype? This would be the hypothesis.

2. Figure 3. If $n=2$, you should not show the SEM, but the standard deviation and possibly the individual points from the 2 experiments. It also looks like FF1220 has impaired PRC1 binding from Fig 3A and C. So it may be good to nuance the result, as this mutant could affect chromosome condensation state but also to some extent the central spindle.

3. The pink arrows refer to "Arrows mark the weak metaphase spindle staining for KIF4A FF1220AA, and the cortical spread of microtubules in the FF1154AA PRC1-binding mutant in anaphase." in the figure legend. However this is not discussed in the text and it is not quantified or examined in other supporting experiments. So should be removed as well as related figure legend.

4. Are there other sites on Kif4a phosphorylated by Aurora kinases?

It is not clear the sites are Aurora A or B specific, as the figure 7 does not show that. There still seems to be some staining in Figure D after treatment with Aurora inhibitors. The chromosomes are not stained. Punctate staining is present instead. Are these kinetochores?

Minor points:

Fig 6D How is chromosome congression/unaligned quantified? The chromokinesin depletion often leads to less compact metaphase, but not necessarily means misaligned. Fig 6E. The difference between time to anaphase between WT and S799A/T801A mutant is small. Is it the same with a kinesin mutant with disrupted ATPase activity? Can you confirm it in another assay/other way? How many repeats in these experiments? Not sure what error bars are. Please show standard deviation.

Figure 1 should be in supplementary figures, it does not add anything new to manuscript.

Figure 6A and B don't also show anything new (phenotype studied in Wandke et al, 2012) and should be in supplementary figures.

The number of times the experiments is repeated is not clear. In one place, it says the experiment was done twice. Could this be stated for each experiment.

List the RNAi sequence for Kid

Kif4a S799A/T801A is not a phosphorylation mutant but rather a non-phosphorylatable mutant

The staining of Kif4a pT799 is not similar to that of Kif4a. There seem to be some staining remaining after inhibition with both Aurora A and B inhibitor. Stain with other marker.

Reviewer #2 (Comments to the Authors (Required)):

Chromokinesin KIF4 localizes both to the chromosomes and to the spindle midzone during cell division. It has been established that physical interactions with different partners control the subcellular targeting of KIF: the chromosome localization depends on the interaction with condensin I, a key molecule for mitotic chromatin architecture, while the localization to the spindle midzone requires the interaction with PRC1, a key mitotic microtubule organizer that preferentially bundles microtubule in an anti-parallel manner at the spindle midzone. In this manuscript, the authors tried to dissect the multiple molecular functions of KIF4 by designing mutants specifically defective for the interaction with condensin I or with PRC1. After confirming the specific influences of the mutations on localization to the chromosomes and the midzone (Figs. 2 and 3), they examined PRC1 localization (Fig. 4 A, B & D), anaphase spindle elongation (Fig. 4C), chromosome condensation (Fig. 5). In Fig. 6 and 7, the role of the interaction with condensin-I was further detailed by comparing the phenotypes of KIF4 condensin-I interaction mutants with the perturbation of phosphorylation by Aurora kinases and depletion of another chromokinesin, KID. Based on these, the authors discuss the cross talk between the KIF4 targeting and distinct activity gradients of Aurora kinases.

While the multifunctionality of KIF4 makes this molecule particularly interesting among many mitotic factors, it makes understanding of the molecular function of KIF4 and its regulation by mitotic kinases difficult. For example, although mutual dependencies for the midzone localization between KIF4 and PRC1 and their cooperation in the organization of the antiparallel microtubule overlaps have been well accepted through *in vivo* and *in vitro* works, whether and how their interaction plays a specific role in this process remained unclear. This is because the KIF4 mutants used as being defective for interaction with PRC1 lacked a large C-terminal sequence (bigger than ~120 residues) (Bieling 2010; Hu, C.-K., Coughlin, M., Field, C.M., and Mitchison, T.J. (2011). KIF4 regulates midzone length during cytokinesis. *Curr. Biol.* 21, 815-824. Although the latter has not been mentioned in the current manuscript, it is highly important for this manuscript. See below for more details). A large deletion might have affected not only the interactions with PRC1 and with condensin I but also with other factors such as microtubules (eg. the basic patch might electrostatically strengthen the interaction of KIF4 with microtubules via an interaction with the acidic tubulin tails). Separation of functions by designing function-specific point mutations such as reported in this manuscript has been long awaited. Thus, this manuscript is potentially suitable for the *Journal of Cell Biology*. However, there are several points to be addressed before publication.

Phenotypes of FF1154AA cells

The FF1154AA mutant cells in Fig. 4A, 4D, and S1C haven't formed a furrow yet while the other cells to be compared in the same panels have already started furrowing (This becomes obvious by increasing the brightness of the images). Figs. S1D and E show that furrowing occurs earlier in the FF1154AA mutant than the control. Taking this into account, it is likely that the FF1154AA cells in Fig. 4A, 4D, and S1C are at earlier cell cycle stages than others. By comparing cell shapes (mainly by the degree of furrowing) between the IF images and the Figs. S1D images, we can reasonably estimate that they were at ~2 min after anaphase onset while the WT cells were at ~10 min. It has been reported that the affinity of PRC1 to microtubules and KIF4 are regulated by CDK1 phosphorylation. The broader PRC1 signal in the FF1154AA cells (Fig. 4A and 4B) might be

explained just by the difference in the timing of fixation relative to the entry into anaphase. In Fig. 4D, the diameter of the cell at the furrow region in KIF4A WT is significantly smaller than that of the equatorial diameter of the FF1154AA cells. It is unclear whether the more compact radial distribution of PRC1 in the WT cell in the right panel (90° rotation) is reflecting the genuine difference in the PRC1 and spindle dynamics or whether it's caused by constriction of the cell equator (PRC1 can't be outside of the cell membrane).

Statistical analysis of the broadening of PRC1 localization by the FF1544AA mutation is missing. The authors' standard or rationale for choosing anaphase cells for comparison between cell lines is unclear. The intensity, length along the spindle axis and radial distribution of PRC1 signals in sufficiently large numbers of the cells should be quantified at multiple anaphase stages determined by a reasonable timing marker. For example, the cells can be fixed at multiple time points after the release from nocodazole (or a proteasome inhibitor) and analyzed based on the degree of chromatin decondensation. Live imaging can be a superior alternative.

Comparison with Hu 2011

Although this has not been referenced in this manuscript, Hu et al 2011 (PMID: 21565503) examined the role of KIF4-PRC1 interaction in the regulation of the length of the midzone microtubule overlaps and the spindle elongation in anaphase. They compared a KIF4 mutant lacking C-terminal ~130 residues with the wild type in their abilities to suppress the broad PRC1 localization caused by KIF4 depletion. Hu et al concluded that, contrary to the expectation and the current results shown in Fig. 4, PRC1 focusing was rescued by the tailless, PRC1-binding defective mutant. The reason for this difference is unclear. It could be due to the difference in the construct (a large deletion vs a point mutation), efficiency of RNAi-depletion, expression method of the rescue constructs, or the timing of observation (Hu et al. 2011 seem to have observed the cells at slightly later stages). The existence of a preceding work by Hu et al 2011 must have been recognized by the corresponding author since this paper has been cited multiple times in previous publications from his group. The current authors should make a stronger case for their conclusions, properly discussing the preceding work.

KIF4A pull-down assay

In Fig. 3, the signals of PRC1 in the input of FF1154AA are weaker than the corresponding bands for the other cells. Is this just a matter of exposure in western blotting, or reflecting the lower PRC1 levels in this cell line? If this was due to variable exposure, how would this influence the image of KIF4A IP lanes? What would a side-by-side comparison on the same membrane look like?

Metaphase localization of the condensin-binding mutants

The mutants defective for condensin I-binding shows localization to the metaphase spindle. Does this depend on the interaction with PRC1 or its own motor domain? Is the PRC1 localization during metaphase affected by this mutation? What would the phenotype of FF1154AA and FF1220AA double mutant cells look like?

Cyclin B blots

The images of cyclin B blots for FF154AA and FF1220AA cell lines seem to be identical although there are differences in the exact regions cropped.

Influences of the inhibition of Aurora kinases

In explaining Fig. 7D, the authors emphasize that the signal on the equatorial chromosomes was not completely eliminated by the Aurora A inhibitor (MLN8237) ("while the signal on equatorial chromosomes was retained"). This is true but the actual images of pT779 staining give an

impression that the signal on the equatorial chromosomes is also weaker in the MLN-treated cell than in the control cell. In contrast, although the authors describe "Inhibition of Aurora B resulted in loss of the equatorial signal for pT799", the pT779 signals are clearly "retained". Roughly speaking, both the inhibitors seem to affect both the pT779 signals on the polar and equatorial chromosomes more or less. More quantitative analyses with counter-stain markers for centromere and chromosome arms should be performed.

Introduction

page 3 "Aurora A also phosphorylates and inhibits the centromere-associated kinesin CENP-E involved in efficient chromosome congression (Kapoor et al., 2006; Kim et al., 2010). A phosphorylation site mutant form of CENP-E cannot be inhibited by Aurora A and thus traps non-bioriented chromosomes at the spindle poles (Kapoor et al., 2006; Kim et al., 2010)." This is confusing as it reads that active CENP-E traps non-bioriented chromosomes at the spindle poles. This is counter-intuitive, considering that CENP-E is a driver of chromosome congression. The conclusion in Kim et al. (2010) is that reduced affinity to microtubules (and thus reduced processivity) by Aurora A phosphorylation at the poles promotes the selectivity towards the dense bundles of kinetochore microtubules (K-fiber) than the radial array of astral microtubules.

page 4 "This simple model poses a problem, since non-equatorial chromosomes will be released from bound microtubules by either the action of Aurora A or B, and thus fail to undergo movement towards the cell equator. How is the congression of these non-equatorial chromosomes actively promoted?" is not appropriate as the summary of the previous works in the previous paragraph is not precise, and

page 5 "PRC1 and KIF4A are mutually dependent on each other for localization to the anaphase spindle midzone where they together determine central spindle length (Bieling et al., 2010; Subramanian et al., 2010)." Both of these papers are not appropriate for in vivo functions since both are about the role of cooperation of these factors in in vitro reconstitution of microtubule bundles. The direct interaction between PRC1 and KIF4 was first reported by Kurasawa et al. (2004) and further studied in Zhu, C. et al. (2005) *Mol Biol Cell* 16, 3187-3199, Zhu, C. et al. (2006) *Proc Natl Acad Sci U S A* 103, 6196-6201 and Hu, C.K. et al. (2011) *Curr Biol* 21, 815-824.

- Please add time stamps on each time frame of the movies

Re: JCB manuscript #201905194

Response to the referees

Referee 1

In this paper, the authors examine the distinct roles of Kif4a in mitosis. Kif4a associates with condensin to target to chromosomes and control chromosome packaging, and with PRC1 to target to the midzone and define the midzone region to measure central spindle length. The authors generate Kif4a mutants that associate with either partners to distinguish their function in these two processes driven by different interaction partners. They identify the motif responsible for the interaction with PRC1 (FF1154), while the other has previously been reported (FF1220). They then test the function of the mutants in chromosome morphology and condensation and in central spindle length. The role of Kif4a in these processes is known from previous studies (mainly RNAi). However, these site-directed mutants are much better tools to probe the specific function of Kif4a and provide the basis for elegant experiments, as shown here. The FF1220 seem to not interact very efficiently with PRC1 however so this motif may be important for interaction with condensin and PRC1.

The first part of the study uses the known functions of KIF4A to characterise the separation of function mutants used in the second part of the work. As the referee notes, these are “*much better tools*”, since we now have access to point mutants that disrupt specific interactions of KIF4A, rather than large deletions which may alter the function for more than one reason. As a consequence, we are able to extend these previous studies.

To comment on the specific point made by the referee about FF1220AA. KIF4A FF1220AA shows reduced interaction with condensin I and reduced association with the chromosome axis compared to wild type KIF4A. However, it binds to PRC1 and localise to the central spindle similarly to wild type KIF4A (Figures 1, 2 and S1). Importantly, FF1220AA also rescues the spread in PRC1 seen in anaphase cells lacking KIF4A. By contrast, the FF1154AA mutant binds to condensin and the chromosome axis but does not bind to PRC1 or localise to the anaphase spindle, and fails to rescue the spread in PRC1.

The second part of the paper about Aurora -regulation of Kif4a is less extensive in the paper. The authors show the non-phosphorylatable mutant on sites S799/T801 (previously identified, Nunes Basto et al 2013) rescues the chromosome compaction defect (from Kif4a phenotype), but does not rescue the chromosome alignment. These findings would indicate that Aurora kinases control Kif4a-mediated chromosome congression but not chromosome packaging. The authors here don't discuss the regulation of Kif4a at midzone by aurora

kinases (previously published). Overall the paper and data are good, interesting and solid. However there seem to be an imbalance with a large emphasis on Aurora regulation of kinesin-4 in the introduction and discussion of the paper, but a modest contribution of these experimental results, possibly a little incomplete on the topic of regulation. Consolidating figure 1-4 into 2/3 figures would rebalance the emphasis of the paper on Aurora regulation of Kif4a, especially a couple of figures or panels don't show anything new (like Kif4a localization).

We accept the view of the referee that the role of Aurora A in regulation of KIF4A needed to be investigated in greater depth. To strengthen the case that Aurora A regulates KIF4A, we have gone back to an assay originally used by Tarun Kapoor to study metaphase plate formation and chromosome congression in a 2006 Science paper. We have modified this assay to include a transient Aurora A or Aurora B inhibition step (Figure 8A). This modified assay shows a clear dependence on KIF4A for recovery from Aurora A but not Aurora B inhibition. We then extend this approach by complementing the KIF4A depleted cells with different mutants. Consistent with our hypothesis, KIF4A can only rescue chromosome congression if it retains condensin I binding and the Aurora phosphorylation site. This data with quantification of phenotypes is included in Figure 8B-8E.

The paper was restructured to move Figure 1 to a new supplemental Figure S1A-S1B. Additional supporting biochemical analysis was also included as Figure S1C.

While we accept that there is some overlap with previous work, previous studies did not bring together the analysis of KIF4A function at chromosomes and in the different stages of spindle formation in the way we have done here. It is only by careful characterization and analysis of the different KIF4A binding mutants and phosphorylation signals that we have been able to test the hypothesis proposed here.

Major points:

1. The non-phosphorylatable Kif4a mutant rescues the depletion phenotype for compaction, similar to WT Kif4a (Fig 5). However using a phosphoantibody, they show Kif4a on chromosomes is phosphorylated (Fig 7). What happens to the SE mutant? Does it also rescue? Similarly, the authors show S799A/T801A is defective in chromosome alignment. What about the S799E/T801E? Is this one rescuing the phenotype? This would be the hypothesis.

We have added data on the behavior of phospho-mimetic KIF4A. This mutant rescues the Aurora A dependent chromosome congression phenotype in both assays we have used. First, in cells co-depleted of KIF4A and KID (Figure 6E and S4C). Second, in the assay for

Aurora A dependent chromosome congression where cells are depleted of only KIF4A and then rescued with different mutants (Figure 8D and 8E).

2. Figure 3. If n=2, you should not show the SEM, but the standard deviation and possibly the individual points from the 2 experiments. It also looks like FF1220 has impaired PRC1 binding from Fig 3A and C. So it may be good to nuance the result, as this mutant could affect chromosome condensation state but also to some extent the central spindle.

We changed the SEM to standard deviation on these graphs.

3. The pink arrows refer to "Arrows mark the weak metaphase spindle staining for KIF4A FF1220AA, and the cortical spread of microtubules in the FF1154AA PRC1-binding mutant in anaphase." in the figure legend. However this is not discussed in the text and it is not quantified or examined in other supporting experiments. So should be removed as well as related figure legend.

The main text now refers to these arrows. We feel that it is important to point out these features of the staining rather than ignore it. It becomes relevant in the context of anaphase spindle organisation for these same mutants (Figure 4A).

4. Are there other sites on Kif4a phosphorylated by Aurora kinases?

It is not clear the sites are Aurora A or B specific, as the figure 7 does not show that. There still seems to be some staining in Figure D after treatment with Aurora inhibitors. The chromosomes are not stained. Punctate staining is present instead. Are these kinetochores?

Phosphorylation of KIF4A in the C-terminal tail by CDK1 has been suggested in the literature (Poonpern et al. 2017, Hertz et al. 2016). We have previously mapped Aurora sites to T799 and S801. In vitro kinase assays comparing the phosphorylation of KIF4A and the KIF4A T799A/S801A point mutant show a strongly reduced level of phosphorylation in the mutant (see Bastos et al., 2013). This supports the view that T799 and S801 are the major Aurora sites in KIF4A.

Analysis of the localization of phosphorylated KIF4A and requirement for Aurora A in generating the polar signal has been improved. We revisited these experiments and titrated the inhibitors to achieve a greater degree of specificity (Figure 7D). In addition, we added marker staining for the centromere protein CENP-A as requested (Figure 7E). It is important to note that we don't propose that KIF4A along the entire chromosome arms becomes phosphorylated, only the region that overlaps with the polar gradient of Aurora A, and possibly the centromeric pool of Aurora B. The punctate signal commented on by the

referee, is reduced by Aurora B but not Aurora A inhibition. Furthermore, it overlaps with the CENP-A marker. Together, these two observations are consistent with the idea that the punctate signal it is in the centromeric region of the chromosome.

Minor points:

Fig 6D How is chromosome congression/unaligned quantified? The chromokinesin depletion often leads to less compact metaphase, but not necessarily means misaligned. Fig 6E. The difference between time to anaphase between WT and S799A/T801A mutant is small. Is it the same with a kinesin mutant with disrupted ATPase activity? Can you confirm it in another assay/other way? How many repeats in these experiments? Not sure what error bars are. Please show standard deviation.

Cells with are grouped into one of two categories for this analysis: either those with a metaphase plate or those with unaligned chromosomes. This is then plotted in the graphs with mean and standard deviation indicated. The figure legends have been updated to include information for cell numbers counted and number of experiments for all conditions. Statistical tests are also described and significance indicated on the graphs as appropriate.

Figure 1 should be in supplementary figures, it does not add anything new to manuscript. Figure 6A and B don't also show anything new (phenotype studied in Wandke et al, 2012) and should be in supplementary figures.

We have followed the advice of the referee and made this change.

The number of times the experiments is repeated is not clear. In one place, it says the experiment was done twice. Could this be stated for each experiment.

The figure legends have been updated to ensure that they all include information of cell numbers counted and number of experiments for all conditions.

List the RNAi sequence for Kid

RNA duplex sequence are now listed in the Materials and Methods section.

Referee 2

Chromokinesin KIF4 localizes both to the chromosomes and to the spindle midzone during cell division. It has been established that physical interactions with different partners control the subcellular targeting of KIF: the chromosome localization depends on the interaction with condensin I, a key molecule for mitotic chromatin architecture, while the localization to the spindle midzone requires the interaction with PRC1, a key mitotic microtubule organizer that preferentially bundles microtubule in an anti-parallel manner at the spindle midzone. In this manuscript, the authors tried to dissect the multiple molecular functions of KIF4 by designing mutants specifically defective for the interaction with condensin I or with PRC1. After confirming the specific influences of the mutations on localization to the chromosomes and the midzone (Figs. 2 and 3), they examined PRC1 localization (Fig. 4 A, B & D), anaphase spindle elongation (Fig. 4C), chromosome condensation (Fig. 5). In Fig. 6 and 7, the role of the interaction with condensin-I was further detailed by comparing the phenotypes of KIF4 condensin-I interaction mutants with the perturbation of phosphorylation by Aurora kinases and depletion of another chromokinesin, KID. Based on these, the authors discuss the cross talk between the KIF4 targeting and distinct activity gradients of Aurora kinases.

*While the multifunctionality of KIF4 makes this molecule particularly interesting among many mitotic factors, it makes understanding of the molecular function of KIF4 and its regulation by mitotic kinases difficult. For example, although mutual dependencies for the midzone localization between KIF4 and PRC1 and their cooperation in the organization of the antiparallel microtubule overlaps have been well accepted through in vivo and in vitro works, whether and how their interaction plays a specific role in this process remained unclear. This is because the KIF4 mutants used as being defective for interaction with PRC1 lacked a large C-terminal sequence (bigger than ~120 residues) (Bieling 2010; Hu, C.-K., Coughlin, M., Field, C.M., and Mitchison, T.J. (2011). KIF4 regulates midzone length during cytokinesis. *Curr. Biol.* 21, 815-824. Although the latter has not been mentioned in the current manuscript, it is highly important for this manuscript. See below for more details). A large deletion might have affected not only the interactions with PRC1 and with condensin I but also with other factors such as microtubules (eg. the basic patch might electrostatically strengthen the interaction of KIF4 with microtubules via an interaction with the acidic tubulin tails). Separation of functions by designing function-specific point mutations such as reported in this manuscript has been long awaited. Thus, this manuscript is potentially suitable for the *Journal of Cell Biology*. However, there are several points to be addressed before publication.*

Phenotypes of FF1154AA cells

The FF1154AA mutant cells in Fig. 4A, 4D, and S1C haven't formed a furrow yet while the other cells to be compared in the same panels have already started furrowing (This becomes obvious by increasing the brightness of the images). Figs. S1D and E show that furrowing occurs earlier in the FF1154AA mutant than the control. Taking this into account, it is likely that the FF1154AA cells in Fig. 4A, 4D, and S1C are at earlier cell cycle stages than others. By comparing cell shapes (mainly by the degree of furrowing) between the IF images and the Figs. S1D images, we can reasonably estimate that they were at ~2 min after anaphase onset while the WT cells were at ~10 min.

Cells were carefully selected for equivalent time in anaphase. We have added a live cell imaging panel (Figure 3C) which shows that at ~2min of anaphase, chromosomes are only just separated. None of the images we show in the other figures are at this early stage. As we demonstrate in Figure S2C the FF1154AA cells start to furrow earlier at ~8min of anaphase compared to ~10min for control cells. Thus, the central spindle starts to become more compacted, not less. However, at the time points we show (~8min), the central spindle has spread out further towards the cell cortex in the FF1154AA cells when compared to KIF4A WT. In fact, we think this spread is what triggers the premature furrowing. Taken together, these observations support the hypothesis that the KIF4A-PRC1 interaction is important for maintaining the radial organisation of microtubules and collecting bundles away from the cell cortex.

It has been reported that the affinity of PRC1 to microtubules and KIF4 are regulated by CDK1 phosphorylation. The broader PRC1 signal in the FF1154AA cells (Fig. 4A and 4B) might be explained just by the difference in the timing of fixation relative to the entry into anaphase. In Fig. 4D, the diameter of the cell at the furrow region in KIF4A WT is significantly smaller than that of the equatorial diameter of the FF1154AA cells. It is unclear whether the more compact radial distribution of PRC1 in the WT cell in the right panel (90{degree sign} rotation) is reflecting the genuine difference in the PRC1 and spindle dynamics or whether it's caused by constriction of the cell equator (PRC1 can't be outside of the cell membrane).

This point is addressed by the responses given immediately above and below.

Statistical analysis of the broadening of PRC1 localization by the FF1544AA mutation is missing. The authors' standard or rationale for choosing anaphase cells for comparison between cell lines is unclear. The intensity, length along the spindle axis and radial distribution of PRC1 signals in sufficiently large numbers of the cells should be quantified at multiple anaphase stages determined by a reasonable timing marker. For example, the cells

can be fixed at multiple time points after the release from nocodazole (or a proteasome inhibitor) and analyzed based on the degree of chromatin decondensation. Live imaging can be a superior alternative.

We agree that it is difficult to make precise conclusions about time from images of fixed cells, especially when comparing mutant phenotypes. To do this we have staged the cells according to the extent of chromosome separation in anaphase, choosing a point at the end of anaphase A/start of anaphase B. Live cell imaging of PRC1 has therefore been added to address concerns about the precise timing of its recruitment to the anaphase spindle. Fixed cell images are equivalent to 6-8min after the metaphase-anaphase transition. We have previously published similar analysis in the Journal of Cell Biology for the control situation, and now add the analysis of the KIF4A FF1154AA PRC1-binding mutant to this (see Figure 3C). The result agrees with the data collected on fixed cells shown in other figures. Other relevant live cell imaging data on the localization of KIF4A, central spindle length and time of furrowing is shown in Figure 2D and S2C-S2D.

Error bars have been added to the graphed quantification in Figure 3B (n=7-16). These were omitted in the original figure. As explained above, this is equivalent to 6-8min after the metaphase-anaphase transition. Carrying out this measurement at multiple timepoints would not be informative. Early on there is little PRC1 to measure. At later timepoints the cell starts to furrow compressing the signal, and this has altered timing and become irregular/asymmetric in the different mutants. For these reasons we use ~6-8min after the metaphase-anaphase transition

Comparison with Hu 2011

Although this has not been referenced in this manuscript, Hu et al 2011 (PMID: 21565503) examined the role of KIF4-PRC1 interaction in the regulation of the length of the midzone microtubule overlaps and the spindle elongation in anaphase. They compared a KIF4 mutant lacking C-terminal ~130 residues with the wild type in their abilities to suppress the broad PRC1 localization caused by KIF4 depletion. Hu et al concluded that, contrary to the expectation and the current results shown in Fig. 4, PRC1 focusing was rescued by the tailless, PRC1-binding defective mutant. The reason for this difference is unclear. It could be due to the difference in the construct (a large deletion vs a point mutation), efficiency of RNAi-depletion, expression method of the rescue constructs, or the timing of observation (Hu et al. 2011 seem to have observed the cells at slightly later stages). The existence of a preceding work by Hu et al 2011 must have been recognized by the corresponding author since this paper has been cited multiple times in previous publications from his group. The

current authors should make a stronger case for their conclusions, properly discussing the preceding work.

We have corrected the text to cite the Hu et al 2011 paper at the appropriate points. Deletion of the C-terminus of KIF4A or point mutations at FF1154 result in a loss of PRC1 interaction, and in our hands loss of central spindle targeting. We have performed this analysis using fixed and live cell imaging, with both GFP and mCherry tagged constructs. The referee notes that Hu et al 2011 look at later time points than we do, and we would add the rescue is not to the level of PRC1 seen in the controls. There may be other simple explanations like siRNA efficiency, but we don't feel it is helpful to speculate about this. In general, our results are in reasonable agreement with the literature described in the introduction showing the KIF4A-PRC1 interaction is important for KIF4A targeting. It is unclear to us how a mutant KIF4A would have an effect on PRC1 if the two proteins cannot interact.

KIF4A pull-down assay

In Fig. 3, the signals of PRC1 in the input of FF1154AA are weaker than the corresponding bands for the other cells. Is this just a matter of exposure in western blotting, or reflecting the lower PRC1 levels in this cell line? If this was due to variable exposure, how would this influence the image of KIF4A IP lanes? What would a side-by-side comparison on the same membrane look like?

The referee makes the point that PRC1 levels appear different in the FF1154AA condition. Endogenous PRC1 is blotted for in the figure (was Figure 3, now Figure 2), so this should be equivalent in the different conditions. We have carefully checked the original data and found an error in the figure where blot exposures were not correctly matched. In addition, a blot was swapped between the FF1154 and FF1220 samples. This has been corrected in the revised figure.

To address the specific comment, we have also provided scans of the original western blots. The level and signal for PRC1 is the same for all the cell lines used and a side-by-side comparison is provided for the KIF4A WT and FF1154AA mutants (see Figure below). Bands are numbered in red; original hand-written labelling on the blots films lists the experimental samples and antibodies used). Two exposures are used in the final figure: a short exposure for the input material and a longer exposure for the IP fraction.

Figure for reviewer. Western blots of KIF4A IP experiments. Top panels: long and short exposures are shown for PRC1 input and IP fractions.

The panels at the bottom show: 1. Short exposure, input PRC1 in KIF4A WT. 2. Long exposure, IP fraction PRC1 in KIF4A WT. 3. Short exposure, input PRC1 in KIF4A FF1154AA IP. 4. Long exposure, IP fraction PRC1 in KIF4A FF1154AA IP.

Metaphase localization of the condensin-binding mutants

The mutants defective for condensin I-binding shows localization to the metaphase spindle. Does this depend on the interaction with PRC1 or its own motor domain? Is the PRC1 localization during metaphase affected by this mutation? What would the phenotype of FF1154AA and FF1220AA double mutant cells look like?

Mutants defective for condensin I-binding show a faint spindle localisation and this may be due to either a weak interaction with PRC1 or some residual motor activity. We don't see strong PRC1 staining on the metaphase spindle (Figure S1A).

The FF1154AA/1220AA double mutant is shown in Figures 1 and 2 with live cell imaging shown in Figure 3. We also provide biochemical analysis in Figure S1C. As expected this mutant fails to bind PRC1 and condensin I, and does not localise to either prometaphase/metaphase chromosome arms or the central spindle.

Cyclin B blots

The images of cyclin B blots for FF154AA and FF1220AA cell lines seem to be identical although there are differences in the exact regions cropped.

This was an error in the figure and has been corrected.

Influences of the inhibition of Aurora kinases

In explaining Fig. 7D, the authors emphasize that the signal on the equatorial chromosomes was not completely eliminated by the Aurora A inhibitor (MLN8237) ("while the signal on equatorial chromosomes was retained"). This is true but the actual images of pT779 staining give an impression that the signal on the equatorial chromosomes is also weaker in the MLN-treated cell than in the control cell. In contrast, although the authors describe "Inhibition of Aurora B resulted in loss of the equatorial signal for pT799", the pT779 signals are clearly "retained". Roughly speaking, both the inhibitors seem to affect both the pT779 signals on the polar and equatorial chromosomes more or less. More quantitative analyses with counter-stain markers for centromere and chromosome arms should be performed.

Reviewer 1 made a similar point. Analysis of the localization of phosphorylated KIF4A and requirement for Aurora A in generating the polar signal has been improved. We revisited these experiments and titrated the inhibitors to achieve a greater degree of specificity (Figure 7D). In addition, we added marker staining for the centromere protein CENP-A as requested (Figure 7E).

September 10, 2019

Re: JCB manuscript #201905194R

Prof. Francis A Barr
University of Oxford
Department of Biochemistry South Parks Road
Oxford OX1 3QU
United Kingdom

Dear Prof. Barr,

Thank you for submitting your revised manuscript entitled "Aurora A promotes chromosome congression by activating the condensin-dependent pool of KIF4A". The manuscript has been seen by the original reviewers whose full comments are appended below. While the reviewers continue to be overall positive about the work in terms of its suitability for JCB, some important issues remain.

You will see that reviewer #1 continues to feel that better imaging data is necessary to support the idea that T799-phosphorylated Kif4a localizes to the kinetochore and further feels that this data needs to be properly quantified. Reviewer #2 has also noted a few lingering issues with the paper. In particular, this reviewer feels that the comparison of the radial distribution of PRC1 and MLKP1 in the WT and mutant cells (figure 4) is not adequate since the WT cells have already formed a cleavage furrow (and the mutant cells have not) and thus the distribution of these proteins are necessarily more restricted since the equatorial diameter of the WT cell is smaller than that of the mutant. The reviewer has also provided a helpful visual to illustrate what s/he means (attached to this email).

We hope that you will be able to address each of these remaining reviewer concerns in a final revised version of the paper. Please note, though, that we do not agree with reviewer #2 that the radial compaction data should be removed from the paper.

As you know, our general policy is that papers are considered through only one revision cycle; however, given that the suggested changes are relatively minor we are open to one additional short round of revision. Please note that I will expect to make a final decision without additional reviewer input upon resubmission.

Please submit the final revision within one to two months, along with a cover letter that includes a point by point response to the remaining reviewer comments.

Thank you for this interesting contribution to Journal of Cell Biology. You can contact me or the scientific editor listed below at the journal office with any questions, cellbio@rockefeller.edu or call (212) 327-8588.

Sincerely,

Arshad Desai, PhD
Monitoring Editor
JCB

Tim Spencer, PhD
Interregnum Executive Editor
Journal of Cell Biology

Reviewer #1 (Comments to the Authors (Required)):

Overall, my comments well have been addressed except point 4.

Can the authors stain with CENP-A (mouse commercially available) or ACA (human antibody, commercially available) AND pT799 in the same image so we can see if the staining is at kinetochores. It is a very easy experiment to do.

Now the authors say:

"Together, these two observations are consistent with the idea that the punctate signal it is in the centromeric region of the chromosome."

If they can show it instead of say it is an idea, it would be better. Right now it is tempting to think but because the staining is punctate, it's hard to know if it is actually staining kinetochores.

Also figure 7 needs to be quantified to show changes in phosphorylation levels at aligned and misaligned chromosomes.

Reviewer #2 (Comments to the Authors (Required)):

The majority of my points have been addressed in this revision. However, the most crucial one, the comparison between the WT and FF1154AA cells in Figure 4 remains inappropriate. As pointed out in my original comment, this is due to

1. the problem in staging fixed cells by the extent of the chromosome separation
2. comparison between the WT cells that have already formed a cleavage furrow and the mutant cells that have not yet formed the furrow
3. lack of statistical analysis

These led to a major concern that the differences in PRC1 and MKLP1 localization patterns between the WT and FF1154AA cells in Figure 4 might not be so obvious if the WT cells in comparable cell shape (i.e. before furrowing) were compared with the mutant cells. The radial distributions of PRC1 and MKLP1 in both the WT and FF1154AA cells in this figure might just be limited simply by the radius of the cell equator (not by the spindle defects due to the mutation). This concern has not been properly addressed. None of the three problems listed above has been resolved. The newly added data from live recordings (Fig. 3C) rather support this concern since the radial expansion of PRC1 in the FF1154AA cell is not observed at any time points as clearly as in Fig. 4. Moreover, the large variation in the kinetics of chromosome separation among the same FF1154AA mutant cells (Fig. 3C vs Fig. S2C) further strengthens the concern about using the extent of chromosome segregation as a measure for the timing after anaphase onset. Thus, the revised manuscript is not yet suitable for publication in its current form (if Fig. 4 is to be included as it is).

Having said the above, I think that the authors' hypothesis on the role of KIF4A-PRC1 interaction in the radial organization of microtubules is indeed a plausible and interesting one that is worth examining more properly. Indeed, a close look at the images presented in Fig. 3C detects slight

differences in the radius of the main disk-like distribution of PRC1 bundles and the patterns of the flanking weaker signals that failed to be compacted into the main assembly. I think that the authors should remove the data in Fig. 4 as well as the discussion about the radial compaction (this is of secondary importance in comparison with the functional dissection of binding partners although doing so would slightly discount the novelty of this manuscript in the context of cytokinesis), or perform a more quantitative and statistical analysis of the live cell imaging of PRC1 (or MKLP1, microtubules etc.) for a more unambiguous and reliable conclusion.

Please find my point-by-point response to the authors' rebuttal below (the authors response in ---" followed by my comments)

Phenotypes of FF1154AA cells

---- "Cells were carefully selected for equivalent time in anaphase. We have added a live cell imaging panel (Figure 3C) which shows that at ~2min of anaphase, chromosomes are only just separated. None of the images we show in the other figures are at this early stage."

In this particular example in Fig. 3C, the chromosome separation is indeed minimum at 2 min of anaphase. However, in Fig. S2C, the FF1154AA cell shows much more extensive chromosome separation at 2 min after anaphase onset, which is comparable to that at 6~8 min in the control. These observations are clear indications that the extent of chromosome separation can't be a reliable marker for the progression of anaphase (in this type of experiment where the spindle architecture is affected).

---- "As we demonstrate in Figure S2C the FF1154AA cells start to furrow earlier at ~8min of anaphase compared to ~10min for control cells."

This is not precise. The FF1154AA cell in Fig. S2C shows the sign of furrow ingression (i.e. the appearance of a negative curvature at the equatorial region of the outline of the cell) at 4 min. At 8 min the FF1544AA cell has formed a deep furrow (the furrow diameter is less than 50% of that of 2 min in Fig. S2C). According to Fig. S2D, the mean (or median? where is the legend?) for the mutant is less than 7 min and, at 8 min, only 5/19 WT cells showed furrowing while all the FF1154AA cells except one showed furrowing.

---- "Thus, the central spindle starts to become more compacted, not less."

This is true if the WT and FF1154AA cells were compared at the same time after anaphase onset. The problem is how this is guaranteed.

---- "However, at the time points we show (~8min), the central spindle has spread out further towards the cell cortex in the FF1154AA cells when compared to KIF4A WT."

The key problem here is that it is impossible to determine the timing after anaphase onset simply by the morphologies (chromosome separation and cell shape etc.) of the fixed cells. If we agree that the furrowing occurs earlier in FF1154AA cells than in WT cells, to compare the morphologies of the central spindle at the same timing after anaphase onset, the extent of the furrowing in the mutant cells should be more than that of the wild type counterpart. However, both in Figure 4A and 4B, FF1154AA cells with shallower or no furrowing were presented.

---- "In fact, we think this spread is what triggers the premature furrowing. Taken together, these observations support the hypothesis that the KIF4A-PRC1 interaction is important for maintaining the radial organization of microtubules and collecting bundles away from the cell cortex." Honestly, I agree that these are plausible ideas and are worth examining and I even believe that they are true. However, the presented data are not sufficiently rigorous.

---- "We agree that it is difficult to make precise conclusions about time from images of fixed cells, especially when comparing mutant phenotypes. To do this we have staged the cells according to the extent of chromosome separation in anaphase, choosing a point at the end of anaphase A/start of anaphase B."

As described above, chromosome separation can't be a reliable measure for the timing after anaphase onset. The FF1154AA mutant promotes spindle elongation and thus can also accelerate chromosome separation. If the cells were chosen based on the extent of chromosome separation, mutant cells at earlier timing after anaphase onset would tend to be chosen.

---- "Live cell imaging of PRC1 has therefore been added to address concerns about the precise timing of its recruitment to the anaphase spindle. Fixed cell images are equivalent to 6-8min after the metaphase-anaphase transition. We have previously published similar analysis in the Journal of Cell Biology for the control situation, and now add the analysis of the KIF4A FF1154AA PRC1-binding mutant to this (see Figure 3C)."

The efforts for the additional data are highly appreciated. However,...

---- "The result agrees with the data collected on fixed cells shown in other figures. Other relevant live cell imaging data on the localization of KIF4A, central spindle length and time of furrowing is shown in Figure 2D and S2C-S2D."

This is not true. The FF1154AA cell in Fig. 3C indeed shows less sharp localization of PRC1 along the spindle axis, which is consistent with the data by immunofluorescence and with previous reports on KIF4 depletion. On the other hand, however, it is not obvious whether there is a difference in the radial distribution of PRC1 between the WT and FF1154AA cells in this panel. None of the stills from Fig. 3C PRC1 clearly shows such a radially expanded localization of PRC1 as distinctive as in Fig. 4A. It is not obvious for the readers how representative these movies and images in Fig. 4 are. Moreover, there is a huge difference in the patterns of chromosome segregation between the FF1154AA cell in Fig. 3C and the one in Fig. S2C, re-emphasizing the question of whether the extent of chromosome separation is a suitable standard for staging the fixed cells.

---- "Error bars have been added to the graphed quantification in Figure 3B (n=7-16). These were omitted in the original figure."

This is good. However, it is just about the PRC1 distribution along the spindle axis. The statistical analysis is still missing for the radial distribution of PRC1 or MKLP1.

---- "As explained above, this is equivalent to 6-8min after the metaphase-anaphase transition." This doesn't fit very well with the authors' own data in Fig. S2D. According to this graph, all except one FF1154AA cells (the exact number is unclear (n=14?)) because the points in this group are overlapping each other) have started furrowing by 8 min after anaphase onset while only 5 in 19 WT cells have done so. In Fig. 4A and 4B, in contrast, both the FF1154AA cells have NOT formed the furrow while both the WT cells have. The chance that this occurs with an unbiased choice would be very low.

---- "Carrying out this measurement at multiple time points would not be informative. Early on there is little PRC1 to measure. At later time points the cell starts to furrow compressing the signal, and this has altered timing and become irregular/asymmetric in the different mutants. For these reasons we use ~6-8min after the metaphase-anaphase transition."

These are sort of reasonable arguments and confirm the importance of the comparison between different cell types at the precisely equivalent timing. The PRC1 distribution in the WT cells in Fig. 3C indicates that the radial compaction is still underway in this timing. Thus, more careful analysis

would be necessary to obtain a firm conclusion.

EP Abraham Professor of Mechanistic Cell Biology
Professor Francis Barr

South Parks Road
Oxford OX1 3QU
www.bioch.ox.ac.uk
Tel: +44 (0)1865 613212
Fax: +44 (0)1865 613213
Email: head@bioch.ox.ac.uk

10th October 2019

Re: JCB manuscript #201905194 revision

Title: Aurora A promotes chromosome congression by activating the condensin-dependent pool of KIF4A

Dear Dr Desai

We greatly appreciate your patience and the decision to allow a final revision of the manuscript. The comments and figure provided by referee #2 was extremely helpful, and we have given some careful thought to how best to address the issue raised. I can only apologize for failing to understand what reviewer #2 meant in their initial comments. Our solution has been to revisit the time lapse imaging approach already used for Figure 3B, and extend the analysis of these cells with some extra experimental work. This data enables us to define early and late stages for both the KIF4A WT and FF1154AA mutant. We have then used this information to select the fixed cells shown in Figure 4 where we examine radial organization of PRC1 and other central spindle proteins. Rather than showing single time points in Figure 4, which we accept was not ideal, we now show cells prior to, and during furrowing. Using the time lapse data as a calibration tool addresses the issue raised by referee #1, although we accept that timings are never going to be absolute with an approach like this. However, we feel that this does allow us to define early and late stages pre/post furrow initiation as shown. The text on pages 9-11 and legends to Figures 3 and 4 have been revised to explain these changes and the new data.

Reviewer #1 asked for co-staining in Figure 7 with a centromere/kinetochore marker and quantification of the inhibitor treatments. We have included staining for CENP-A in Figure 7E and quantification in Figure 7F.

We hope these changes are sufficient to ensure publication of our work in the Journal of Cell Biology. Thank you for the time you have spent as monitoring editor, and please thank the reviewers for the really valuable comments.

Yours sincerely, Francis Barr

Reviewer #1 (Comments to the Authors (Required)):

Overall, my comments will have been addressed except point 4.

Can the authors stain with CENP-A (mouse commercially available) or ACA (human antibody, commercially available) AND pT799 in the same image so we can see if the staining is at kinetochores. It is a very easy experiment to do.

Reviewer #1 asked for co-staining in Figure 7 with a centromere/kinetochore marker and quantification of the inhibitor treatments. We have included staining for CENP-A in Figure 7E and quantification in Figure 7F.

Reviewer #2 (Comments to the Authors (Required)):

The majority of my points have been addressed in this revision. However, the most crucial one, the comparison between the WT and FF1154AA cells in Figure 4 remains inappropriate. As pointed out in my original comment, this is due to

1. the problem in staging fixed cells by the extent of the chromosome separation
2. comparison between the WT cells that have already formed a cleavage furrow and the mutant cells that have not yet formed the furrow.

The comments and figure provided were extremely helpful, and we have given some careful thought to how best to address the issue raised in points 1 and 2. I can only apologize for failing to understand what was meant in the initial comments. Our solution has been to revisit the time lapse imaging approach already used for Figure 3B, and extend the analysis of these cells with some extra experimental work. This data enables us to define early and late stages for both the KIF4A WT and FF1154AA mutant. We have then used this information to select the fixed cells shown in Figure 4 where we examine radial organization of PRC1 and other central spindle proteins. Rather than showing single time points in Figure 4, which we accept was not ideal, we now show cells prior to, and during furrowing. Using the time lapse data as a calibration tool addresses the issue raised, although we accept that timings are never going to be absolute with an approach like this. However, we feel that this does allow us to define early and late stages pre/post furrow initiation as shown. We now determine furrow timing in both the KIF4A WT and FF1154A conditions using time lapse imaging (Figure 3B-3F). This is marked on the images and shown in graphed form.

The text on pages 9-11 and legends to Figures 3 and 4 have been revised to explain these changes and refer to the new data.

3. lack of statistical analysis

As part of the revisions we have included additional analysis of protein distribution.

These led to a major concern that the differences in PRC1 and MKLP1 localization patterns between the WT and FF1154AA cells in Figure 4 might not be so obvious if the WT cells in comparable cell shape (i.e. before furrowing) were compared with the mutant cells. The radial distributions of PRC1 and MKLP1 in both the WT and FF1154AA cells in this figure might just be limited simply by the radius of the cell equator (not by the spindle defects due to the mutation). This concern has not been properly addressed. None of the three problems listed above has been resolved. The newly added data from live recordings (Fig. 3C) rather support this concern since the radial expansion of PRC1 in the FF1154AA cell is not observed at any time points as clearly as in Fig. 4. Moreover, the large variation in the kinetics of chromosome separation among the same FF1154AA mutant cells (Fig. 3C vs Fig. S2C) further strengthens the concern about using the extent of chromosome segregation as a measure for the timing after anaphase onset. Thus, the revised manuscript is not yet suitable for publication in its current form (if Fig. 4 is to be included as it is).

Having said the above, I think that the authors' hypothesis on the role of KIF4A-PRC1 interaction in the radial organization of microtubules is indeed a plausible and interesting one that is worth examining more properly. Indeed, a close look at the images presented in Fig. 3C detects slight differences in the radius of the main disk-like distribution of PRC1 bundles and the patterns of the flanking weaker signals that failed to be compacted into the main assembly. I think that the authors should remove the data in Fig. 4 as well as the discussion about the radial compaction (this is of

EP Abraham Professor of Mechanistic Cell Biology
Professor Francis Barr

secondary importance in comparison with the functional dissection of binding partners although doing so would slightly discount the novelty of this manuscript in the context of cytokinesis), or perform a more quantitative and statistical analysis of the live cell imaging of PRC1 (or MKLP1, microtubules etc.) for a more unambiguous and reliable conclusion.

Please find my point-by-point response to the authors' rebuttal below (the authors response in ---" " followed by my comments)

Phenotypes of FF1154AA cells

---- "Cells were carefully selected for equivalent time in anaphase. We have added a live cell imaging panel (Figure 3C) which shows that at ~2min of anaphase, chromosomes are only just separated. None of the images we show in the other figures are at this early stage."

In this particular example in Fig. 3C, the chromosome separation is indeed minimum at 2 min of anaphase. However, in Fig. S2C, the FF1154AA cell shows much more extensive chromosome separation at 2 min after anaphase onset, which is comparable to that at 6-8 min in the control. These observations are clear indications that the extent of chromosome separation can't be a reliable marker for the progression of anaphase (in this type of experiment where the spindle architecture is affected).

We now use the live cell imaging and combination of chromosome separation and cell furrowing to create individual "calibration" curves for KIF4A WT and FF1154AA to stage the cells. In addition, we show cells at both pre and post-furrow initiation in Figure 4.

---- "As we demonstrate in Figure S2C the FF1154AA cells start to furrow earlier at ~8min of anaphase compared to ~10min for control cells."

This is not precise. The FF1154AA cell in Fig. S2C shows the sign of furrow ingression (i.e. the appearance of a negative curvature at the equatorial region of the outline of the cell) at 4 min. At 8 min the FF1154AA cell has formed a deep furrow (the furrow diameter is less than 50% of that of 2 min in Fig. S2C). According to Fig. S2D, the mean (or median? where is the legend?) for the mutant is less than 7 min and, at 8 min, only 5/19 WT cells showed furrowing while all the FF1154AA cells except one showed furrowing.

---- "Thus, the central spindle starts to become more compacted, not less."

This is true if the WT and FF1154AA cells were compared at the same time after anaphase onset. The problem is how this is guaranteed.

As explained above, the use of live cell imaging to create individual "calibration" curves for KIF4A WT and FF1154AA enables us to stage the cells more accurately. We also show two time points to highlight the difference between early and late stages.

---- "However, at the time points we show (~8min), the central spindle has spread out further towards the cell cortex in the FF1154AA cells when compared to KIF4A WT."

The key problem here is that it is impossible to determine the timing after anaphase onset simply by the morphologies (chromosome separation and cell shape etc.) of the fixed cells. If we agree that the furrowing occurs earlier in FF1154AA cells than in WT cells, to compare the morphologies of the central spindle at the same timing after anaphase onset, the extent of the furrowing in the mutant cells should be more than that of the wild type counterpart. However, both in Figure 4A and 4B, FF1154AA cells with shallower or no furrowing were presented.

We agree that it is impossible to determine the timing after anaphase onset simply by chromosome separation and cell shape of the fixed cells. This is why we have used the live cell imaging (Figure 3) to support Figure 4. In the revised Figure 4, cells are pre and post-furrowing stages are shown.

---- "In fact, we think this spread is what triggers the premature furrowing. Taken together, these observations support

the hypothesis that the KIF4A-PRC1 interaction is important for maintaining the radial organization of microtubules and collecting bundles away from the cell cortex."

Honestly, I agree that these are plausible ideas and are worth examining and I even believe that they are true. However, the presented data are not sufficiently rigorous.

We measure radial organization of PRC1 in KIF4A WT and FF1154AA cells in Figure 3E and 3F, and correlate this to furrow initiation time. This supports the view we put forward in the text. We should also note that this is not a major aspect of our study, but does support the view the FF1154AA mutant is defective for the anaphase specific function of KIF4A relating to PRC1.

---- "We agree that it is difficult to make precise conclusions about time from images of fixed cells, especially when comparing mutant phenotypes. To do this we have staged the cells according to the extent of chromosome separation in anaphase, choosing a point at the end of anaphase A/start of anaphase B."

As described above, chromosome separation can't be a reliable measure for the timing after anaphase onset. The FF1154AA mutant promotes spindle elongation and thus can also accelerate chromosome separation. If the cells were chosen based on the extent of chromosome separation, mutant cells at earlier timing after anaphase onset would tend to be chosen.

In the revised manuscript we have used the live cell imaging (Figure 3) to support Figure 4. In the revised Figure 4, cells are pre and post-furrowing stages are now shown.

---- "Live cell imaging of PRC1 has therefore been added to address concerns about the precise timing of its recruitment to the anaphase spindle. Fixed cell images are equivalent to 6-8min after the metaphase-anaphase transition. We have previously published similar analysis in the Journal of Cell Biology for the control situation, and now add the analysis of the KIF4A FF1154AA PRC1-binding mutant to this (see Figure 3C)."

The efforts for the additional data are highly appreciated. However,...

---- "The result agrees with the data collected on fixed cells shown in other figures. Other relevant live cell imaging data on the localization of KIF4A, central spindle length and time of furrowing is shown in Figure 2D and S2C-S2D."

This is not true. The FF1154AA cell in Fig. 3C indeed shows less sharp localization of PRC1 along the spindle axis, which is consistent with the data by immunofluorescence and with previous reports on KIF4 depletion. On the other hand, however, it is not obvious whether there is a difference in the radial distribution of PRC1 between the WT and FF1154AA cells in this panel. None of the stills from Fig. 3C PRC1 clearly shows such a radially expanded localization of PRC1 as distinctive as in Fig. 4A. It is not obvious for the readers how representative these movies and images in Fig. 4 are. Moreover, there is a huge difference in the patterns of chromosome segregation between the FF1154AA cell in Fig. 3C and the one in Fig. S2C, re-emphasizing the question of whether the extent of chromosome separation is a suitable standard for staging the fixed cells.

As explained above we have fully revised Figures 3 and 4 to address this point.

---- "Error bars have been added to the graphed quantification in Figure 3B (n=7-16). These were omitted in the original figure."

This is good. However, it is just about the PRC1 distribution along the spindle axis. The statistical analysis is still missing for the radial distribution of PRC1 or MKLP1.

We have moved this graph to Figure S1E, and added a complete new set of measurements of PRC1 distribution at multiple time points for both KIF4A WT and FF1154AA in Figure 7E and 7F. The radial distributions as a function of time are clearly different, and fall within the cell cortex in both cases.

---- "As explained above, this is equivalent to 6-8min after the metaphase-anaphase transition."

This doesn't fit very well with the authors' own data in Fig. S2D. According to this graph, all except one FF1154AA cells (the exact number is unclear (n=14?) because the points in this group are overlapping each other) have started

EP Abraham Professor of Mechanistic Cell Biology
Professor Francis Barr

furrowing by 8 min after anaphase onset while only 5 in 19 WT cells have done so. In Fig. 4A and 4B, in contrast, both the FF1154AA cells have NOT formed the furrow while both the WT cells have. The chance that this occurs with an unbiased choice would be very low.

The cell lines in Figure S2D express KIF4A WT or FF1154AA but not GFP-PRC1 and cannot be directly compared to those in the main figure in terms of precise timing. However, the major phenotypic changes observed are the same: increased chromosome separation and premature cytokinesis. This is provided as supporting data and shows the effects are not due to overexpression of the GFP-PRC1 construct.

---- "Carrying out this measurement at multiple time points would not be informative. Early on there is little PRC1 to measure. At later time points the cell starts to furrow compressing the signal, and this has altered timing and become irregular/asymmetric in the different mutants. For these reasons we use ~6-8min after the metaphase-anaphase transition."

These are sort of reasonable arguments and confirm the importance of the comparison between different cell types at the precisely equivalent timing. The PRC1 distribution in the WT cells in Fig. 3C indicates that the radial compaction is still underway in this timing. Thus, more careful analysis would be necessary to obtain a firm conclusion.

We have added a more extensive analysis of furrow timing Figure 3B-3F using measurements at multiple time points. At very early stages there is too little PRC1 to measure, so we start this at 4min then point cells switch from anaphase A to anaphase B. This supports the view that there are clear differences in furrow timing in KIF4A WT and FF1154AA cells.

October 24, 2019

RE: JCB Manuscript #201905194RR

Prof. Francis A Barr
University of Oxford
Department of Biochemistry South Parks Road
Oxford OX1 3QU
United Kingdom

Dear Francis:

Thank you for submitting your revised manuscript entitled "Aurora A promotes chromosome congression by activating the condensin-dependent pool of KIF4A". We would be happy to publish your paper in JCB pending final revisions necessary to meet our formatting guidelines (see details below).

A. MANUSCRIPT ORGANIZATION AND FORMATTING:

Full guidelines are available on our Instructions for Authors page, <http://jcb.rupress.org/submission-guidelines#revised>. **Submission of a paper that does not conform to JCB guidelines will delay the acceptance of your manuscript.**

1) Text limits: Character count for Articles and Tools is < 40,000, not including spaces. Count includes title page, abstract, introduction, results, discussion, and acknowledgments. Count does not include materials and methods, figure legends, references, tables, or supplemental legends. At the moment, you are slightly over this limit but we should be able to give you the extra space this time.

2) Figure formatting: Scale bars must be present on all microscopy images, including inset magnifications. Molecular weight or nucleic acid size markers must be included on all gel electrophoresis.

3) Statistical analysis: Error bars on graphic representations of numerical data must be clearly described in the figure legend. The number of independent data points (n) represented in a graph must be indicated in the legend. Statistical methods should be explained in full in the materials and methods. For figures presenting pooled data the statistical measure should be defined in the figure legends. Please also be sure to indicate the statistical tests used in each of your experiments (both in the figure legend itself and in a separate methods section) as well as the parameters of the test (for example, if you ran a t-test, please indicate if it was one- or two-sided, etc.). Also, since you used parametric tests in your study (e.g. t-tests, ANOVA, etc.), you should have first determined whether the data was normally distributed before selecting that test. In the stats section of the methods, please indicate how you tested for normality. If you did not test for normality, you must state something to the effect that "Data distribution was assumed to be normal but this was not

formally tested."

4) Materials and methods: Should be comprehensive and not simply reference a previous publication for details on how an experiment was performed. Please provide full descriptions (at least in brief) in the text for readers who may not have access to referenced manuscripts. The text should not refer to methods "...as previously described."

5) Please be sure to provide the sequences for all of your primers/oligos and RNAi constructs in the materials and methods. You must also indicate in the methods the source, species, and catalog numbers (where appropriate) for all of your antibodies.

6) Microscope image acquisition: The following information must be provided about the acquisition and processing of images:

a. Make and model of microscope

b. Type, magnification, and numerical aperture of the objective lenses

c. Temperature

d. imaging medium

e. Fluorochromes

f. Camera make and model

g. Acquisition software

h. Any software used for image processing subsequent to data acquisition. Please include details and types of operations involved (e.g., type of deconvolution, 3D reconstitutions, surface or volume rendering, gamma adjustments, etc.).

7) References: There is no limit to the number of references cited in a manuscript. References should be cited parenthetically in the text by author and year of publication. Abbreviate the names of journals according to PubMed.

8) Supplemental materials: There are strict limits on the allowable amount of supplemental data. Articles/Tools may have up to 5 supplemental figures. At the moment, you are meet this limit but please bear it in mind when revising.

Please also note that tables, like figures, should be provided as individual, editable files. A summary of all supplemental material should appear at the end of the Materials and methods section.

9) eTOC summary: A ~40-50 word summary that describes the context and significance of the findings for a general readership should be included on the title page. The statement should be written in the present tense and refer to the work in the third person.

10) Conflict of interest statement: JCB requires inclusion of a statement in the acknowledgements regarding competing financial interests. If no competing financial interests exist, please include the following statement: "The authors declare no competing financial interests." If competing interests are declared, please follow your statement of these competing interests with the following statement: "The authors declare no further competing financial interests."

11) ORCID IDs: ORCID IDs are unique identifiers allowing researchers to create a record of their various scholarly contributions in a single place. At resubmission of your final files, please consider providing an ORCID ID for as many contributing authors as possible.

B. FINAL FILES:

-- High-resolution figure and video files: See our detailed guidelines for preparing your production-ready images, <http://jcb.rupress.org/fig-vid-guidelines>.

Thank you for your attention to these final processing requirements. Please revise and format the manuscript and upload materials within 7-14 days.

Thank you for this interesting contribution, we look forward to publishing your paper in Journal of Cell Biology.

Sincerely,

Arshad Desai
Monitoring Editor
Journal of Cell Biology

Tim Spencer, PhD
Interregnum Executive Editor
Journal of Cell Biology